# GROUND-TRUTH SUBGRAPHS FOR BETTER TRAINING AND EVALUATION OF KNOWLEDGE GRAPH AUGMENTED LLMS

## ABSTRACT

Retrieval of information from graph-structured knowledge bases represents a promising direction for improving the factuality of LLMs. While various solutions have been proposed, a comparison of methods is difficult due to the lack of challenging QA datasets with ground-truth targets for graph retrieval. We present SynthKGQA, a LLM-powered framework for generating high-quality Knowledge Graph Question Answering datasets from any Knowledge Graph, providing the full set of ground-truth facts in the KG to reason over questions. We show how, in addition to enabling more informative benchmarking of KG retrievers, the data produced with SynthKGQA also allows us to train better models. We apply SynthKGQA to Wikidata to generate GTSQA, a new dataset designed to test zero-shot generalization abilities of KG retrievers with respect to unseen graph structures and relation types, and benchmark popular solutions for KG-augmented LLMs on it.

## 1 INTRODUCTION

Despite significant advances over the years, Large Language Models (LLMs) are still unreliable when asked to provide factual information, as hallucinations (LLMs outputting plausible but wrong answers) remain one of the central problems for LLM applications (Huang et al., 2025). The predominant solution to improving LLM trustworthiness is Retrieval Augmented Generation (RAG), where information pertinent to the query is retrieved from a corpus of knowledge and added to the prompt (Lewis et al., 2020; Borgeaud et al., 2022; Izacard et al., 2023). While RAG traditionally retrieves information from documents, another important use case is retrieval from graph structured repositories such as Knowledge Graphs (KGs). KGs encode facts as (subject, predicate, object) triples and are a highly efficient solution to store *relational* information while also being easier to maintain, update, and fact-check compared to textual documents.

KG-augmented LLMs are evaluated on Knowledge Graph Question Answering (KGQA) benchmarks, where supporting facts need to be extracted from the KG and combined to produce the answer. While many KGQA datasets have appeared over the years (see Peng et al. (2024) for a recent survey), little attention has been paid to their quality, reliability and limitations. The concurrent study by Zhang et al. (2025) has estimated the degree of factual correctness of the questions in widely-used benchmarks, such as WebQSP (Yih et al., 2016), CWQ (Talmor & Berant, 2018) and GrailQA (Gu et al., 2021), to be only between 30 and 60%. Moreover, several benchmarks are no longer challenging for SOTA KG-augmented LLMs and are now close to being saturated, limiting their usefulness. Even for datasets like ComplexWebQuestions (CWQ), which include multi-hop questions, it is impossible to provide an actual measure of question complexity due to the lack of *ground-truth answer subgraphs*, the golden targets for retrieval. This also means that KG-RAG retrievers cannot be evaluated on their own, but only end-to-end, by looking at the final answer provided by the LLM – which, since different solutions use different LLMs and prompting schemes, results in noisy comparisons. Moreover, for KG retrievers that require training, the ground-truth answer subgraph is indispensable to provide supervision signal: using existing datasets, it has to be approximated with the set of shortest paths from seed entities to answer nodes. As we show in Section 6, this is often a bad approximation, which penalizes the final quality of the retriever. Furthermore, most KGQA benchmarks are from the previous decade and based on the now discontinued Freebase KG (Bollacker et al., 2008), thus containing outdated facts and answers that might be in conflict with the more recent information

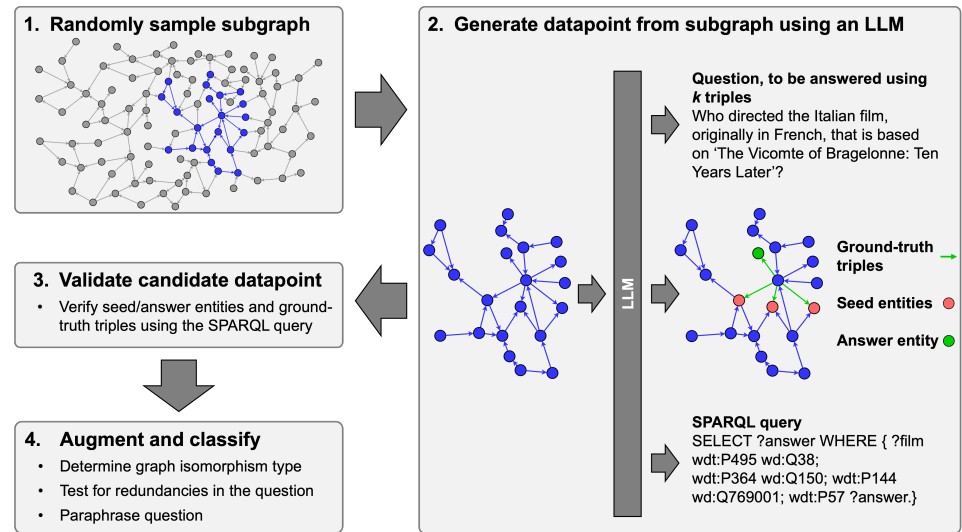

Figure 1: The steps performed by SynthKGQA to generate and validate questions and ground-truth subgraphs from an arbitrary Knowledge Graph.

available to LLMs. Moreover, these datasets have likely been seen during pretraining of recent LLMs (Ding et al., 2024), leading to data leakage. Question variety, in terms of KG relation and entity types used for the grounding facts, is also typically limited, as questions are either manually (through crowd-sourcing) or procedurally (through logical query manipulation) generated from predefined query templates, often resulting in unnatural, or ambiguously phrased, natural language questions (see Zhang et al. (2025) for a more detailed discussion of limitations and pitfalls).

We propose to tackle all these challenges by leveraging the advanced abilities of frontier LLMs to remove the need for a human in the loop in the creation of KGQA datasets. Our contributions are:

- SynthKGQA[1], a new framework (Figure 1) for generating large synthetic KGQA datasets from any KG, which provides high-quality, diverse questions with procedurally-verified ground-truth answer subgraphs and SPARQL queries, allowing to easily update the dataset whenever the underlying KG is modified.

- *Ground-Truth Subgraphs for Question Answering* (GTSQA)[1], a challenging new dataset with 32,099 questions spanning 27 different structures for the ground-truth answer subgraph, grounded in the regularly-updated Wikidata KG (Vrandečić & Krötzsch, 2014), and specifically designed to test generalization abilities of KG-RAG models.

- A comprehensive benchmark of SOTA LLMs and KG-augmented LLMs on GTSQA, providing new insights on retrieval abilities and limitations of these models on different structures for the answer subgraph, and different levels of zero-shot generalization.

- An extensive analysis of the benefits of using the ground-truth answer subgraph, instead of shortest paths from seed to answer nodes, as supervision signal for training KG retrievers.

## 2 RELATED WORK

Traditionally, models for KGQA have been evaluated on datasets built using a combination of handcrafted question and SPARQL templates and human annotation, often from the Freebase KG (Bollacker et al., 2008). WebQuestions (Berant et al., 2013) used autocompletions from the Google Suggest API on Freebase entity labels, with answers provided by human annotators – a process resulting prevalently in 1-hop questions. Many datasets are derived from WebQuestions, by filtering and refining the dataset (WebQSP (Yih et al., 2016)) or extending/combining the SPARQL queries to obtain more complex questions (CWQ Talmor & Berant (2018)). More structured, and partially

---

[1]See supplementary material; links to github repository and dataset to be added after double-blind review.

automated, pipelines started to appear with LC-QuAD (Trivedi et al., 2017), mapping logical queries to natural language using question templates and crowd-sourced paraphrasing. Moreover, after Freebase was discontinued in 2016, new datasets shifted to using different KGs, mainly DBpedia (e.g., QALD-9 (Usbeck et al., 2018)) and Wikidata (e.g., LC-QuAD 2.0 (Dubey et al., 2019)), with parallel efforts to migrate to these KGs previous datasets based on Freebase (Azmy et al., 2018). New interest developed also in testing generalization abilities of KGQA models (GrailQA (Gu et al., 2021)) and better quantifying the complexity of questions (GrailQA++ (Dutt et al., 2023)). Only very recently Dammu et al. (2025) and Zhang et al. (2025) have started exploring the use of LLMs in the construction of KGQA datasets. Despite the implementation of validation pipelines to detect and discard bad data, these approaches can still suffer from hallucinations leading to incorrect question-answer pairs (Zhang et al., 2025). Also, crucial information such as the complete set of entities mentioned in each question, or fine-grained measures of question complexity or generalization abilities required to answer, are not fully tracked. A detailed comparison of these LLM-enabled approaches to the framework proposed in this paper can be found in Appendix E.

## 3 THE SYNTHKGQA FRAMEWORK

**Preliminaries**  KGs store facts as labeled directed edges between *entities* in a set $\mathcal{E}$ (the nodes/vertices of the KG), where edge labels are drawn from a set $\mathcal{R}$ of predicates, or *relation types*. Thus, an edge is represented as a subject-predicate-object triple $(h, r, t)$, with $h, t \in \mathcal{E}$ and $r \in \mathcal{R}$; we denote by $\mathcal{T}$ the set of triples in the KG. KGQA questions require retrieving a subgraph $\mathcal{G} \subset \mathcal{T}$ and reason over it to produce an answer. We assume that the answer(s) to a question $q$ are entities in a set $\mathcal{A} \subset \mathcal{E}$. We also assume that, together with the question, the set of *seed entities* $\mathcal{S} \subset \mathcal{E}$ is provided, i.e., the set of entities that are explicitly mentioned in the question. Extracting seed entities from natural language questions or text is an orthogonal task referred to as Named Entity Recognition, or Entity Linking, which is in itself an object of extensive research (Alam et al., 2022; Keraghel et al., 2024).

**SynthKGQA**  We outline SynthKGQA, our proposed framework, which can be applied to any KG in order to construct high-quality synthetic data for KGQA The data generation pipeline comprises of four steps (Figure 1; more details are provided in Appendix A).

1. **Seed subgraph sampling.** We randomly sample (Algorithm 1) a connected subgraph $\mathcal{Q} \subset \mathcal{T}$, containing a few tens of edges, to be used as context for generating the question.

2. **LLM-powered candidate proposal.** The subgraph $\mathcal{Q}$ is passed to the LLM, which is tasked (through few-shot prompting, see Appendix A.1) to generate:
   - a question $q$ that can be answered with $k$ (number specified in the prompt) triples in $\mathcal{Q}$;
   - the list of triples in $\mathcal{Q}$ required to reason over the question, i.e., the *ground-truth answer subgraph* $\mathcal{G} \subset \mathcal{Q} \subset \mathcal{T}$;
   - the answer to the question, which is required to be an entity $a \in \mathcal{E}$ appearing in $\mathcal{G}$;
   - the list of entities explicitly mentioned in the question, i.e., the seed entities $\mathcal{S} \subset \mathcal{E}$, which also need to appear in $\mathcal{G}$;
   - the SPARQL query $l_q$ which encodes the natural-language question $q$ in logical form.

   Note that we use SPARQL as RDF query language, since the Wikidata Query Service[2] is based on it, but our pipeline can be adapted with minimal changes to use any other query language, if working with knowledge bases that have different querying interfaces.

3. **Candidate validation.** The query $l_q$ is executed against the KG to retrieve the full set of answers $\mathcal{A}$; the candidate datapoint is discarded if $\mathcal{A}$ does not contain the answer $a$ provided by the LLM. Similarly, it is discarded if any of the triples that the LLM considers necessary for answering the question, or any of the proposed seed entities, are not in the *full answer subgraph*, obtained by running the associated CONSTRUCT SPARQL query (i.e., the union of all triples in the KG realizing the query template, taking into account all valid substitutions of the variables in $l_q$; for more details, see the example in Appendix A.4).

4. **Augmentation and classification**. In order to increase the diversity of the generated data, we ask a different LLM to paraphrase $q$ in more natural terms. Moreover, by using the

---

[2]https://query.wikidata.org/

Table 1: Overall view of GTSQA.

|  | Train | Test | All |
|---|---|---|---|
| # questions | 30,477 | 1622 | 32,099 |
| # unique relation types | 200 | 362 | 368 |
| # unique entities | 64,435 | 5665 | 68,520 |
| # unique graph isomorphisms | 19 | 14 | 27 |
| # questions with redundant info | 7852 | 0 | 7852 |
| avg # seed entities | 1.65 | 1.92 | 1.66 |
| avg # hops | 1.48 | 2.02 | 1.5 |
| avg # answers | 1.54 | 1.28 | 1.53 |
| avg # ground-truth edges | 2.13 | 3.03 | 2.17 |

ground-truth answer subgraph and SPARQL query, we can provide additional information on each valid datapoint.

- *Graph isomorphism type*: we classify the structure of the graph $\mathcal{G}$ up to isomorphism (see Appendix A.2), as a measure of question complexity. This also allows the user to perform additional filtering, by discarding questions where the answer subgraph does not comply with desired logical requirements (e.g., disconnected graphs; graphs containing loops or hanging branches, not terminating in a seed entity; graphs where the answer node is also used as a seed node, giving rise to self-answering questions).

- *Redundancy*: we check whether the question contains *redundant* information, i.e., if it can be answered by using only a subset $\mathcal{S}' \subsetneq \mathcal{S}$ of the seed entities (see Appendix A.3).

## 4 THE GTSQA DATASET

Using the SynthKGQA framework and GPT-4.1 (OpenAI, 2023) as LLM, we construct *Ground-Truth Subgraphs for Question Answering* (GTSQA), a synthetic KGQA dataset grounded in the Wikidata KG (Vrandečić & Krötzsch, 2014), with $30,477$ questions in the train set and $1622$ questions in the test set. Statistics on the dataset are summarized in Table 1, with more details provided in Appendix B.

GTSQA supports retrieval either from the full Wikidata, or from ogbl-wikikg2 (Hu et al., 2020), which was employed to sample the seed graphs $\mathcal{Q}$. We construct questions where the ground-truth answer subgraph contains at most 6 edges, as we observe that – beyond this point – questions tend to become overly-convoluted and unnatural to formulate in natural language. We also require the ground-truth answer subgraph to be a tree, i.e., connected and acyclic, with all leaves being seed entities or the answer node. The dataset covers 27 different graph isomorphism types for the ground-truth answer subgraph, with questions involving up to 5 hops and up to 5 different seed entities. Full statistics on the connectivity patterns of seed and answer nodes, and relative frequencies in the dataset, are provided in Table A1 and Appendix B. As shown in Figure B2, approximately 70% of the test questions are multi-seed (two or more seed entities) and $67.9\%$ are multi-hop (with $30.8\%$ requiring $\geq 3$ hops). This variety of graph types makes GTSQA much more challenging than commonly-used benchmarks for KGQA (only $34.5\%$ of the test questions in WebQSP (Yih et al., 2016) require more than 1 hop, and none require more than 2). Moreover, the test set of GTSQA has the unique property of containing only non-redundant questions (as defined in Section 3), where the full ground-truth answer subgraph is necessary to answer, which ensures that the provided classification of graph isomorphism types is truly reflective of the complexity of the questions.

A notable feature of GTSQA is that the train-test split is designed to test zero-shot generalization abilities of KG retrievers (in a spirit similar to the construction of GrailQA (Gu et al., 2021) and GrailQA++ (Dutt et al., 2023)). We ensure that the answer nodes for all test questions never appear as an answer in training questions. Moreover, $47.3\%$ of the answer subgraphs in the test set have as isomorphism type one of 8 classes that are not included in the train set (see Table A1), while $37.2\%$ of test questions require reasoning over edges whose relation type is not seen in the train set (see Figure B1 for the distribution of relation types). We also make sure that, for each graph isomorphism type in the test set, the applicable categories (in-distribution, unseen graph type, unseen relation

type) always contain at least 45 different questions, to enable a statistically meaningful study of KG retrievers' performance at this unprecedented granularity level (as we do in Section 5).

We perform a final filtering step on the test set of GTSQA to ensure its value and reliability as a benchmark for KG retrievers: we retain only questions that can be answered correctly by an advanced LLM (GPT-4o-mini) when augmenting the prompt with the ground-truth answer subgraph provided in the dataset, in a consistent way (two successes out of two tries). This is to sanity check that the ground-truth answer subgraph is indeed a valid golden target for retrieval, providing grounding to any insights we extract from the benchmarks. We found that only 0.47% of the generated data failed this test (and was therefore removed), which confirms the very high quality of the synthetic data produced by SynthKGQA (compared, for instance, to KGQAGen-10k (Zhang et al., 2025), where this failure rate is reported at 10.38% with the more capable GPT-4o, and failure causes remain unclear).

## 5 BENCHMARKS

We benchmark SOTA KG-RAG models on GTSQA. As one of the key features of our dataset is that it provides ground-truth answer subgraphs for each question, we can evaluate not just the quality of the models' final answer, but also the quality of the retrieved subgraph, for different graph isomorphism types and generalization abilities required by the question.

**Experimental setting** We consider a variety of models for QA, divided into four categories (specifications in Appendix C.2). 1) *LLM-only*: as a baseline, we take frontier commercial LLMs without any external augmentation pipelines, such as GPT-4.1, GPT-4o-mini (OpenAI, 2023), GPT-5-mini (OpenAI, 2025), Ministral-8B-Instruct, Mistral-Large-2.1 (Mistral AI, 2024) and LLaMA-3.1-8B-Instruct (AI@Meta, 2024). 2) *KG agents*: training-free models using out-of-the-box LLMs to explore the KG starting from the seed entities. They decide on their own, over multiple steps, what neighbors to explore and when to stop. As representatives, we consider Think-on-Graph (ToG; Sun et al. (2024)) and Plan-on-Graph (PoG; Chen et al. (2024)). 3) *Path-based retrievers*: models trained to predict the paths originating from the seed entities and leading to the answer node. Examples are SR (Zhang et al., 2022), Reasoning on Graphs (RoG; Luo et al. (2024)) and Graph-Constrained Reasoning (GCR; Luo et al. (2025)). 4) *All-at-once retrievers*: models trained to score all edges in a large neighborhood of the seed entities in a single pass, and then retrieve the most relevant ones. We consider SubgraphRAG (Li et al., 2025) as a representative.

For all models in categories 2), 3), 4), we use the same LLM (GPT-4o-mini, which – as explained in Section 4 – is able to answer all test questions when provided with the ground-truth answer subgraph) to perform the final reasoning on the retrieved subgraph, in order to make the benchmark as fair as possible and place the focus on the quality of the retrieved subgraph. We use ogbl-wikikg2 as KG for retrieval, to reduce computational complexity (see also Appendix C.1). As common in KGQA benchmarking (Sun et al., 2024; Luo et al., 2024), we evaluate quality of model responses by reporting on Hits (at least one correct answer predicted) and Recall of correct answers, using Exact Match (EM) between the labels of answer nodes and the model output. Moreover, as made possible by GTSQA, we analyze Recall, Precision and F1 score of ground-truth triples in the subgraph retrieved by the model, in addition to Hits and Recall of answer nodes.

**Results** A detailed comparison of the performance of different models on GTSQA is provided in Table 2. Pure LLMs show limited accuracy with an EM Hits score of at most 33.97, confirming the challenging nature of this new benchmark for models that solely rely on their internal knowledge. Even SOTA KG-RAG models – while performing better than pure LLMs – struggle with the task. Trainable subgraph retrievers outperform KG agents, highlighting the importance of fine-tuning models on the target KG. Among trainable retrievers, SubgraphRAG (with 200 retrieved triples) achieves the best results, surpassing all path-based retrievers, which tend to retrieve smaller subgraphs. The F1 score of retrieved triples is low ($< 30\%$) across all retrievers, but our results suggest that increasing the recall of ground-truth triples is more beneficial than increasing precision, especially when the final reasoning is performed by more capable LLMs, better at dealing with long context, as we show in the comparison in Figure C7. For more compact LLMs, however, retrieval precision should not be entirely disregarded. Also note that the recall of ground-truth triples is a much stronger predictor of final model accuracy than the recall of answer nodes (Figure C6), proving the value of working with datasets like GTSQA that provide the full ground-truth answer subgraph.

Table 2: Benchmark of LLMs and KG-RAG models on GTSQA.

| Category | Model | EM | | Ground-truth triples | | | Answer nodes | | |
|---|---|---|---|---|---|---|---|---|---|
| | | Hits | Recall | Recall | Precision | F1 | Hits | Recall | # triples |
| LLM-only | Ministral-8B-Instruct | 10.73 | 10.16 | - | - | - | - | - | - |
| | LLama-3.1-8B-Instruct | 17.11 | 16.33 | - | - | - | - | - | - |
| | GPT-4o-mini | 20.90 | 19.93 | - | - | - | - | - | - |
| | Mistral-Large-2.1 | 23.61 | 22.85 | - | - | - | - | - | - |
| | GPT-5-mini | 31.44 | 30.20 | - | - | - | - | - | - |
| | GPT-4.1 | 33.97 | 32.83 | - | - | - | - | - | - |
| KG agent | PoG | 32.92 | 31.60 | 31.95 | 27.52 | 27.03 | 31.81 | 30.54 | 7.09 |
| | ToG | 45.68 | 44.47 | 35.50 | 6.45 | 10.50 | 41.55 | 40.97 | 9.17 |
| Path-based | SR | 40.63 | 39.10 | 30.22 | 3.44 | 5.69 | 50.25 | 49.39 | 72.94 |
| | GCR | 49.91 | 48.25 | 40.71 | 27.21 | 29.82 | 47.11 | 45.54 | 6.54 |
| | RoG | 57.58 | 55.78 | 54.69 | 24.04 | 27.00 | 72.91 | 71.84 | 72.31 |
| All-at-once | SubgraphRAG (200) | 61.59 | 58.62 | 79.09 | 1.29 | 2.53 | 85.33 | 84.36 | 199.61 |

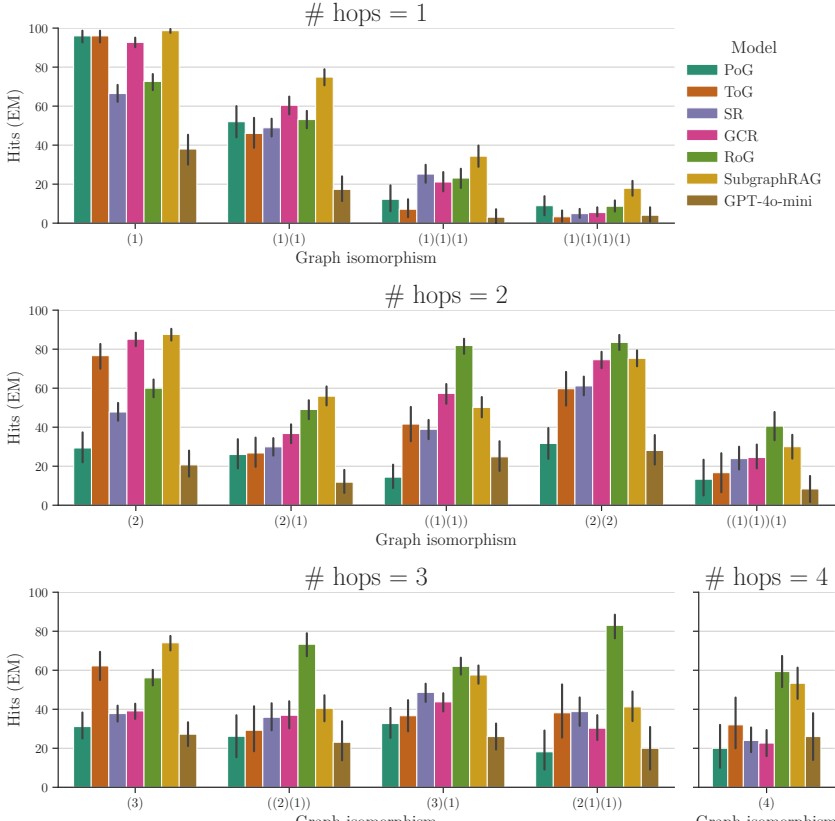

Figure 2: Hits (EM) of KG-RAG models on different graph isomorphism types, compared to the baseline (GPT-4o-mini, no RAG). Isomorphism types are grouped by the maximum number of hops; inside each subplot, moving from left to right corresponds to an increase in the total number of edges in the ground-truth answer subgraph.

Breaking down model performance by graph isomorphism types (Figures 2 and C8), we find that all models, especially KG agents, have very limited accuracy on questions that require intersecting paths from 3 or more seed entities, even if the answer is only one hop away (e.g., graph isomorphisms

(1)(1)(1) and (1)(1)(1)(1)). In this setting, the base LLM on its own also fails consistently. The poor EM is explained by a low recall of ground-truth triples, highlighting a widespread inefficiency in properly expanding and coordinating the search from all seed entities. Indeed, especially for KG agents and path-based methods, the recall of ground-truth triples is on average much lower in multi-seed questions (Figure C8), leading to a worse EM, despite the fact that the answer node is often contained in the retrieved subgraph (Figure C9). This is a consequence of the fact that these models sample paths from distinct seed nodes in a mostly independent way, and therefore struggle with prioritizing common neighbors of the seed nodes that satisfy at the same time all conditions in the query. Unlike prior benchmarks, GTSQA makes this failure mode visible by providing all ground-truth triples and by enforcing that all seed entities are required to answer the question (no redundancy). We provide examples of typical failure cases of different retrievers in Appendix D.

Performance also depends on the number of hops between seed entities and the answer. Comparing KG agents, we observe that PoG matches or outperforms ToG for single-hop questions, but performs severely worse on multi-hop questions, even on simpler graph isomorphism types that involve a single seed entity (e.g., (2) and (3)). This is due to a reduced recall of ground-truth triples (Figure C8). An empirical inspection of the retrieved subgraphs indicates that PoG often prematurely stops the exploration of the KG, over-confidently (and incorrectly, in most cases) believing it has retrieved enough information to answer (see example in Appendix D). We also find a poor performance of GCR, especially when compared to its predecessor RoG, across all graph isomorphism types requiring more than 2 hops from the seed entities. This can mostly be attributed to the implementation restrictions and choices explained in Appendix C.2. Indeed, for these questions we observe that answer node recall drops much more than ground-truth triple recall (Figure C9), because only paths of up to 2 hops from the seed entities can be retrieved. However, GCR also performs worse than RoG on some questions with only 2 hops required, such as graph isomorphisms ((1)(1)) and (2)(1).

Finally, we investigate the zero-shot generalization abilities of trainable retrievers. When a question presents a relation or a graph isomorphism type that has not been seen during training, the improvements over the LLM-baseline are much smaller than for in-distribution questions (Figures 3 and C9). In these settings, we also notice stronger differences between different KG-RAG models. In particular, all path-based retrievers particularly struggle with unseen relation types, even though GCR is much more robust than SR and RoG for this type of generalization, with significantly higher recall of ground-truth triples and EM, as its path predictions are grounded in the KG and hence less dependent on the relation types seen during training. On the other hand, the all-at-once retriever SubgraphRAG maintains strong predictive power on unseen relation types, but underperforms RoG across the majority of unseen graph isomorphism types (Figure C9), with a substantial drop in recall of answer nodes, which, combined with the low precision of retrieved triples typical of this method, results in a poor EM. These fundamental differences between the two classes of retrievers (path-based vs all-at-once) match intuition: path-based methods learn to predict the sequence of relation types that connect seed entities and answer, ignoring the global structure of the answer subgraph, whereas all-at-once retrievers like SubgraphRAG mostly leverage the graph structure during their training. SubgraphRAG especially struggles with graph isomorphism types that require additional projections after an intersection (namely, ((1)(1)), (2)(1)(1)), ((2)(1)), ((1)(1))(1), see Figure C9), a property foreign to the answer subgraphs in the train set – highlighting the lack of robustness of this retriever with respect to new patterns in the logical query (we show an example in Appendix D).

# 6 GROUND-TRUTH SUBGRAPHS TO TRAIN BETTER KG RETRIEVERS

In the absence of ground-truth answer subgraphs in previous datasets, KG retrievers are typically trained using all shortest paths between seed and answer nodes as supervision signal (Zhang et al., 2022; Luo et al., 2024; Li et al., 2025; Luo et al., 2025). The ground-truth subgraphs in GTSQA enable us to conduct experiments addressing the following questions: 1) How much overlap is there between shortest paths and ground-truth subgraphs? 2) Are LLMs still able to answer correctly if augmented with shortest-paths between seed nodes and answer node? 3) Is using ground-truth subgraphs as supervision signal for training KG retrievers better than using shortest paths?

**Q1: Path overlap** Paths of minimal length connecting seed nodes to answer nodes are not guaranteed to be the correct paths required to answer the question. For example, the question "What is the canonization status of Gregory the Illuminator's grandchild?" requires retrieving the 3-hop path

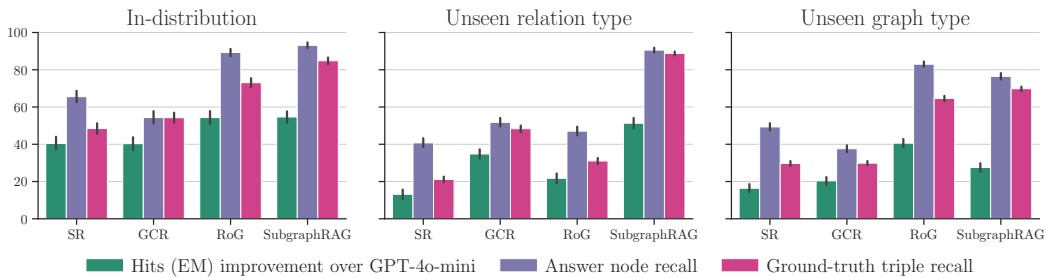

Figure 3: Generalization abilities of trainable KG-RAG models. We measure EM performance in terms of difference with the EM of the baseline (GPT-4o-mini, no RAG).

*Gregory the Illuminator → child → St. Vrtanes I → child → St. Husik I → canonization status → saint*. However, in Wikidata, the shortest path connecting the seed node (*Gregory the Illuminator*) to the answer node (*saint*) consists of a single edge: *Gregory the Illuminator → canonization status → saint*, which clearly provides the wrong reasoning. We refer to similar cases, where the minimal length of paths between seed and answer node is strictly smaller than the length of the path in the ground-truth subgraph, as *shortcuts*. A second source of problems arises from *parallel* (shortest) *paths*: as the distance between the seed and answer node increases, the number of distinct paths of minimal length connecting them is expected to grow exponentially, but only one of them (or none, if there are shortcuts) is a ground-truth path. We provide statistics on the occurrence of shortcuts and parallel paths in GTSQA in Figure B5; their combined effect, as documented in Table B2, strongly reduces the overlap between ground-truth and shortest paths for questions requiring multiple hops. For 4-hop questions, on average, only 13% of the triples along shortest paths are contained in the ground-truth answer subgraph and more than 72% of the ground-truth triples do not lie on paths of minimal length.

**Q2: RAG with shortest paths**   There are cases where following shortcuts or parallel paths, rather than the ground-truth path that we provide in the dataset, can still provide valid reasoning. For instance, the question "Which country is home to the administrative region that includes Nieuw-Weerdinge?", which comes with the 2-hop ground-truth path *Nieuw-Weerdinge → located in the administrative territorial entity → Emmen → country → Netherlands*, can be answered equally well with the shortcut *Nieuw-Weerdinge → country → Netherlands*. Similarly, Wikidata contains pairs of inverse relation types (e.g., *child* and *father*) that give rise to parallel (undirected) paths encoding the same semantical information. We therefore look at the performance of GPT-4o-mini on the test set of GTSQA, when augmenting the prompt with all KG triples on the shortest paths between seed and answer nodes. As shown in Figure 4, there is a strong positive correlation between Hits (EM) and the fraction of ground-truth triples that are contained in the set of shortest path triples. Both metrics degrade sharply with increasing distance between seed and answer node. Note that we already adjust for any spurious effects of the number of hops on EM, since GPT-4o-mini is able to answer all questions correctly when instead we augment the prompt with the triples in the ground-truth answer subgraph. This confirms that shortest paths, even though they end at the correct answer node, often fail to capture the full information required to reason over multi-hop questions.

**Q3: Shortest paths vs ground-truth subgraphs for training**   While the experiments in the previous paragraphs prove that shortest path triples are not as good a target for retrieval as the ground-truth triples provided by SynthKGQA, it remains to understand whether using them as supervision signal for training subgraph retrievers is equally sub-optimal. We consider three retrievers that use path information for training, namely SR (Zhang et al., 2022), RoG (Luo et al., 2024) and SubgraphRAG (Li et al., 2025), train them on GTSQA using either the shortest paths or the ground-truth paths between seed and answer nodes as supervision signal, and then compare the performance on the test set. As reported in Table 3, models trained on the ground-truth subgraphs have EM Hits scores from 5% to 20% higher than their counterparts trained on shortest paths, due to improved recall (up to +27%) and precision (up to +141%) of ground-truth triples in the retrieved subgraphs. The improvements are clear and statistically significant for SubgraphRAG (independently

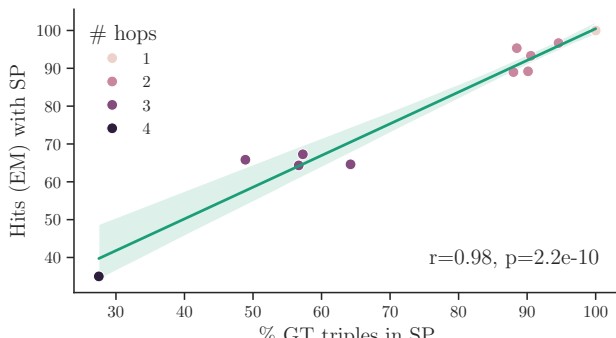

Figure 4: Correlation between the percentage of ground-truth (GT) triples contained in the set of shortest paths (SP) from seed to answer nodes, and EM Hits of GPT-4o-mini when augmented with all SP triples. We display the Pearson correlation coefficient and associated p-value; each dot represents a different isomorphism type of ground-truth answer subgraph in the test set of GTSQA: they are clearly clustered based on the (maximum) number of hops.

Table 3: Improvements in predictive statistics of models trained on ground-truth answer subgraphs (GT), compared to models trained on the shortest paths between seed and answer nodes (SP). For all models, results are averaged over three distinct runs for each of the two training regimes. We also report (between brackets) the metric's % variation and the p-value of the null hypothesis that the mean of distribution of SP scores is larger or equal than the one for GT scores (two-sample, one-sided t-test), showing statistical significance.

| | **Hits (EM)** | | **Recall (GT triples)** | | **Precision (GT triples)** | | **Recall (answer nodes)** | |
| Model | SP | GT | SP | GT | SP | GT | SP | GT |
|---|---|---|---|---|---|---|---|---|
| SR | 33.95 | 40.63 (+20%; 4.5e-12) | 23.74 | 30.22 (+27%; 6.2e-22) | 3.84 | 3.44 (-10%; 0.99) | 36.36 | 49.39 (+36%; 1.0e-39) |
| RoG | 53.08 | 57.58 (+8%; 3.9e-6) | 46.38 | 54.69 (+18%; 4.6e-26) | 10.00 | 24.04 (+141%; 0.0) | 70.68 | 71.84 (+2%; 4.3e-2) |
| SubgraphRAG | 58.47 | 61.59 (+5%; 8.3e-4) | 75.39 | 79.09 (+5%; 1.7e-9) | 1.21 | 1.29 (+6%; 4.0e-7) | 80.46 | 84.36 (+5%; 1.4e-7) |

of the number of retrieved triples, Figure C7), but even more striking for path-based methods. As shown in Figure C10, RoG is the model where the quality of the retrieved subgraph benefits more from eliminating the noise in the training data, with precision increasing by more than $9\times$ for 4-hops questions. Similarly, SR experiences a major boost in answer node recall (more than $2\times$ for 4-hops questions). While for 1-hop questions the differences in all statistics are negligible, the gap sharply increases for questions requiring multiple hops (where, as we've showed, the shortest paths diverge more appreciably from the ground-truth subgraphs).

All this provides definitive evidence on the value of the datasets generated with SynthKGQA not just as benchmarks, but also to train better subgraph retrievers for complex KGQA.

## 7 CONCLUSIONS

In this work, we introduced SynthKGQA, a new framework for constructing large-scale, high-quality synthetic KGQA datasets from arbitrary KGs, with ground-truth answer subgraphs and SPARQL queries for all questions. As concrete application, we released GTSQA, a multi-hop, multi-seed dataset with 30k+ questions based on Wikidata, designed to test how KG-RAG models generalize to unseen answer graph structures and relation types. Our benchmarks show that SOTA KG-RAG models struggle on GTSQA, due to poor retrieval performance that affects especially questions with multiple seed entities. All-at-once retrievers tend to outperform path-based ones and KG agents, due to their higher recall of triples in ground-truth answer subgraphs. Finally, we show how leveraging the SynthKGQA-generated ground-truth subgraphs as supervision signal for training KG retrievers produces better models, with accuracy improvements of up to 30% for multi-hop questions. We anticipate that SynthKGQA and the KGQA datasets generated through it will help benchmark and improve KG retrievers and, consequently, contribute to the development of more trustworthy LLMs.

## REPRODUCIBILITY STATEMENT

The code to run all steps of the SynthKGQA pipeline (detailed in Section 3 and Appendix A) on a custom Knowledge Graph is made available in the supplementary material. We also include the full GTSQA dataset (train and test split) and the code to generate the question-specific graphs for the test questions (Appendix C.1; such graphs will be released together with the dataset). In the supplementary material we also provide all code to run the benchmarks and experiments (Sections 5 and 6) for the evaluated KG-RAG models, whose specifications are detailed in Appendix C.2.

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

# A  ADDITIONAL DETAILS ON SYNTHKGQA

## A.1  DATA GENERATION

We provide additional details on how we query the LLM to generate synthetic data in the SynthKGQA framework (steps 1 and 2 of the pipeline presented in Section 3).

**Seed subgraph**   We first sample a subgraph $\mathcal{Q} \subset \mathcal{T}$ of the KG, which will provide the LLM with the context to generate a question, as asking the LLM to parse the entire KG in its input is unfeasible or impractical. Starting from a random entity $s \in \mathcal{E}$, the subgraph $\mathcal{Q}$ is obtained by progressively expanding a neighborhood around $s$ until the desired size (decided by the user, typically 20-100 edges) is reached. As we aim to construct multi-hop questions, it's important to ensure that the radius of $\mathcal{Q}$ grows quickly, even if the total number of edges in $\mathcal{Q}$ is kept contained. This means prioritizing depth over breadth in the expansion, while maintaining high-degrees of connectivity between nodes, to enable a variety of different isomorphism types for the answer subgraph. Our proposed sampling scheme for $\mathcal{Q}$, used in the construction of GTSQA, is shown in Algorithm 1; we denote by $d(x)$ the degree of a node $x$ (number of edges going in/out of it) and by $\mathcal{N}(x)$ its set of undirected neighbors.

---

**Algorithm 1** Sampling scheme for the seed graph $\mathcal{Q}$

---

**Require:** KG $\mathcal{T} \subset \mathcal{E} \times \mathcal{R} \times \mathcal{E}$; starting node $s \in \mathcal{E}$; node limit $N_{\text{nodes}}$; edge limit $N_{\text{edges}}$
   $N \leftarrow \{s\}$
   $\mathcal{Q} \leftarrow \emptyset$
   **while** $|N| < N_{\text{nodes}}$ and $|\mathcal{Q}| < N_{\text{edges}}$ **do**
      $z \leftarrow \text{choice}(N, p = \text{Softmax}([d(x)^{-1} \text{ for } x \in N]))$       ▷ Sample node to expand
      $M \leftarrow \mathcal{N}(z) \setminus N$       ▷ Retrieve new neighbors of $z$
      **if** $|M| > 0$ **then**
         $n \leftarrow \text{choice}(M, p = \text{Softmax}([d(x)^{-1} \text{ for } x \in M]))$       ▷ Sample neighbor
         $\mathcal{Q} \leftarrow \mathcal{Q} \cup \{(h, r, t) \in \mathcal{T} \text{ s.t. } h = n, t \in N \text{ or } t = n, h \in N\}$   ▷ Add all connections to $n$
         $N \leftarrow N \cup \{n\}$
      **end if**
   **end while**

---

**LLM querying**   The sampled subgraph $\mathcal{Q}$ is given to the LLM to generate one KGQA datapoint. In the prompt we specify the number $k$ of triples in $\mathcal{Q}$ that should be used to reason over the question (i.e., the number of triples in the ground-truth answer subgraph $\mathcal{G}$), and show examples in few-shot prompting style. The RDF identifiers in the knowledge base (e.g., QIDs and PIDs when working with Wikidata) should be included in the prompt, together with labels, for all entities and relations appearing in $\mathcal{Q}$, in order to ensure consistency in the generation of the SPARQL query. An example (for $k = 2$) of the prompt used for the construction of GTSQA is shown below.

---

**LLM Prompt** ($k = 2$)

```
{ "role": "user", "content": "Based on the provided set of knowledge
graph triples, please generate a question that requires combining the
information in exactly 2 of the provided triples for answering.  The
answer should correspond to exactly one node in the provided graph, be
unique and not ambiguous.  Make sure that all 2 of the selected triples
are required for answering the question and that they involve multiple
different entities.  Respond only with
```

- `Question:`  the generated question,

- `Nodes mentioned in the question:`  a semicolon-separated list of the nodes that are explicitly mentioned in the question,

- `Answer:`  the node corresponding to the correct answer with its QID,

- `Triples used:`  a semicolon-separated list of the triples used for answering the question,

- `SPARQL query:`  a SPARQL query to return all answers of the question from the Wikidata knowledge base.

---

```
"},
{ "role": "user", "content": "Graph: Quiaios (Q1024001)-country
(P17)-Portugal (Q45);Savage Nights (Q174785)-filming location
(P915)-Portugal (Q45);Quiaios (Q1024001)-located in the administrative
territory entity (P131)-Figueira da Foz (Q428459);Figueira
da Foz (Q428459)-country (P17)-Portugal (Q45);Savage Nights
(Q174785)-filming location (P915)-Lisbon (Q597);Portugal (Q45)-capital
(P36)-Lisbon (Q597);Lisbon (Q597)-country (P17)-Portugal (Q45);Lisbon
(Q597)-capital of (P1376)-Portugal (Q45);Francisco José Fernandes
Costa (Q357932)-country of citizenship (P27)-Portugal (Q45);Francisco
José Fernandes Costa (Q357932)-place of death (P20)-Figueira da
Foz (Q428459);Francisco José Fernandes Costa (Q357932)-position
held (P39)-Minister of Foreign Affairs (Q4294919);Minister of
Foreign Affairs (Q4294919)-country (P17)-Portugal (Q45);Minister of
Foreign Affairs (Q4294919)-applies to jurisdiction (P1001)-Portugal
(Q45);Francisco José Fernandes Costa (Q357932)-occupation
(P106)-politician (Q82955)
"},
{ "role": "assistant", "content": "
Question:  In which country did Francisco José Fernandes Costa die?,
Nodes mentioned in the question:  Francisco José Fernandes Costa
(Q357932),
Answer:  Portugal (Q45),
Triples used:  Francisco José Fernandes Costa (Q357932)-place of death
(P20)-Figueira da Foz (Q428459); Figueira da Foz (Q428459)-country
(P17)-Portugal (Q45),
SPARQL query:  SELECT ?answer WHERE {wd:Q357932 wdt:P20 ?place.  ?place
wdt:P17 ?answer.}
"},
[...][**OMITTED -- examples with different graph isomorphism types**]
{"role":  "user", "content":  "Graph:  {𝒬}"},
```

## A.2 GRAPH ISOMORPHISMS

We use the notion of *graph isomorphism*, applied to the ground-truth answer subgraph, to provide a simple and objective measure of the complexity of a question in a KGQA dataset. This measure abstracts away the identity of the entities and relation types involved in the answer subgraph and only focuses on the number of seed entities, the number of hops separating them from the answer node and how the paths originating from each seed intersect. In the context of KGQA, this notion first appeared as *reasoning paths* (Das et al., 2022) and *semantic structures* (Li & Ji, 2022), and was then formulated in the same way that we will use it in Dutt et al. (2023) for the construction of GrailQA++.

We say that two questions have the same graph isomorphism type if their ground-truth answer subgraphs $\mathcal{G}$ are isomorphic as labeled graphs, when each node in $\mathcal{G}$ is labeled as "seed" (for seed nodes), "answer" (for the answer node) or "intermediate". This means that there exists a bijection of the sets of vertices of the two graphs that preserves both edges and labels. Note that, while KGs are directed graphs, when computing isomorphism types we consider the answer subgraph as undirected, as KG retrievers should ideally be able to retrieve relevant edges independently of their direction. To provide identifiers for the different graph isomorphism types, the notation based on projections and intersections that is sometimes used in KG reasoning papers (e.g., in Das et al. (2022)) is unfortunately not able to describe complex graph structures without ambiguities. Since, as a data quality requirement, we only consider queries where the answer subgraph is a tree, we adopt a simplified/more readable version of the tree encoding scheme classically used in the AHU algorithm (Aho & Hopcroft, 1974). The answer subgraph is seen as a tree rooted in the answer node, while seed entities are the leaves of the tree. For each leaf, we write $(n)$ if $n$ is the distance of the leaf from its closest branching point (this represents a projection of length $n$ from the seed node). The intersection of paths from multiple seeds at a branching point is denoted by juxtaposition, e.g. $(n)(m)$. Further projections after an intersection point are represented using an additional level of brackets, e.g. $(k(n)(m))$, with $k$ the length of the projection (which can be omitted if $k = 1$). See Table A1 for notations and graphical representations of all the graph isomorphism types appearing in GTSQA.

Table A1: Isomorphism types of ground-truth answer subgraphs in GTSQA and their frequencies in the train and test split. In the graph visualisations, seed nodes are represented in red, while the answer node is marked in green. For the test set, we count separately questions where the answer subgraph is of an isomorphism type not present in the train set (unseen graph type, ugt), questions involving relation types not appearing in the train set (unseen relation type, urt) and questions where the answer subgraph is in-distribution with the train set (id). Percentages are based on the total sizes of the train and test set, respectively.

| Identifier | Visualisation | Dutt et al. (2023) | Train | | Test (ugt) | | Test (urt) | | Test (id) | |
|---|---|---|---|---|---|---|---|---|---|---|
| | | | count | % | count | % | count | % | count | % |
| (1) |  | Iso-0 | 7658 | 25.1 | 0 | 0 | 100 | 6.17 | 50 | 3.08 |
| (2) |  | Iso-1 | 8338 | 27.4 | 0 | 0 | 100 | 6.17 | 50 | 3.08 |
| (1)(1) |  | Iso-2 | 4481 | 14.7 | 0 | 0 | 100 | 6.17 | 50 | 3.08 |
| (2)(1) |  | Iso-3 | 1470 | 4.82 | 0 | 0 | 77 | 4.75 | 50 | 3.08 |
| ((1)(1)) |  | Iso-4 | 0 | 0 | 125 | 7.71 | 0 | 0 | 0 | 0 |
| (3) |  | Iso-5 | 1731 | 5.68 | 0 | 0 | 130 | 8.01 | 50 | 3.08 |
| (2)(2) |  | Iso-6 | 0 | 0 | 139 | 8.57 | 0 | 0 | 0 | 0 |
| (3)(1) |  | Iso-7 | 0 | 0 | 150 | 9.25 | 0 | 0 | 0 | 0 |
| ((2)(1)) |  | Iso-8 | 0 | 0 | 65 | 4.01 | 0 | 0 | 0 | 0 |
| ((1)(1))(1) |  | Iso-9 | 0 | 0 | 60 | 3.7 | 0 | 0 | 0 | 0 |
| (2(1)(1)) |  | Iso-10 | 0 | 0 | 55 | 3.39 | 0 | 0 | 0 | 0 |
| (1)(1)(1) |  | Iso-11 | 5526 | 18.1 | 0 | 0 | 48 | 2.96 | 50 | 3.08 |
| ((1)(1)(1)) |  | Iso-12 | 144 | 0.47 | 0 | 0 | 0 | 0 | 0 | 0 |
| (2)(2)(1) |  | Iso-13 | 24 | 0.08 | 0 | 0 | 0 | 0 | 0 | 0 |
| ((3)(1)) |  | Iso-14 | 1 | 0 | 0 | 0 | 0 | 0 | 0 | 0 |
| (4)(1) |  | Iso-15 | 15 | 0.05 | 0 | 0 | 0 | 0 | 0 | 0 |
| (4) |  | Iso-16 | 0 | 0 | 50 | 3.08 | 0 | 0 | 0 | 0 |
| ((1)(1))(2) |  | Iso-17 | 10 | 0.03 | 0 | 0 | 0 | 0 | 0 | 0 |
| (2)(1)(1) |  | Iso-18 | 907 | 2.98 | 0 | 0 | 0 | 0 | 0 | 0 |
| ((2)(1)(1)) |  | Iso-19 | 3 | 0.01 | 0 | 0 | 0 | 0 | 0 | 0 |
| ((1)(1))(1)(1) |  | Iso-22 | 27 | 0.09 | 0 | 0 | 0 | 0 | 0 | 0 |
| ((2)(1))(1) |  | Iso-23 | 5 | 0.02 | 0 | 0 | 0 | 0 | 0 | 0 |
| ((1)(1)(1))(1) |  | Iso-25 | 18 | 0.06 | 0 | 0 | 0 | 0 | 0 | 0 |
| ((1)(1)(1))(2) |  | Iso-26 | 1 | 0 | 0 | 0 | 0 | 0 | 0 | 0 |
| (1)(1)(1)(1) |  | N/A | 0 | 0 | 123 | 7.58 | 0 | 0 | 0 | 0 |
| (1)(1)(1)(1)(1) |  | N/A | 111 | 0.36 | 0 | 0 | 0 | 0 | 0 | 0 |
| (5) |  | N/A | 7 | 0.02 | 0 | 0 | 0 | 0 | 0 | 0 |

### A.3 REDUNDANCY

As explained in Section 3 (step 3), when filtering the LLM-generated data we check that all seed entities identified by the LLM are used in the SPARQL query. This, however, does not guarantee that all of them are strictly necessary to answer the question, i.e., impose additional constraints compared to the other seeds. A question $q$ contains *redundant* information, if there exists a subset $\mathcal{S}' \subsetneq \mathcal{S}$ such that the SPARQL query $l_{q,\mathcal{S}'}$ corresponding to the subtree $\mathcal{G}_{\mathcal{S}'} \subset \mathcal{G}$ composed by the paths from the seeds in $\mathcal{S}'$ to the answer node $a$ returns the same set of answers $\mathcal{A}$ as the original query $l_q$ (while, in general, we expect the set of answers of $l_{q,\mathcal{S}'}$ to be a strict superset of $\mathcal{A}$). In the case of redundant information, we identify the smallest such $\mathcal{S}'$ (there can be more than one) and classify the graph isomorphism type of $\mathcal{G}_{\mathcal{S}'}$, which should be considered as the "minimal" subgraph(s) sufficient to answer the question. Note that questions with redundant information are still perfectly valid, as all seed entities are mentioned in the question and the information they imply should be therefore retrieved from the KG; however, they present "shortcuts" to the answer that effectively reduce the question complexity, compared to what is measured by the graph isomorphism type of the ground-truth answer subgraph.

For example, the question "Which musical instrument is played by both Yehonatan Geffen's child and Francis Lickerish?" has two seed entities: $\mathcal{S} = \{$Yehonatan Geffen (Q2911403), Francis Lickerish (Q3720616)$\}$. The corresponding SPARQL query

```
SELECT ?answer WHERE { wd:Q2911403 wdt:P40 ?child .
?child wdt:P1303 ?answer. wd:Q3720616 wdt:P1303 ?answer. }
```

has a unique answer in Wikidata, namely *guitar*. The ground-truth answer subgraph contains three edges, (Yehonatan Geffen; child; Aviv Geffen), (Aviv Geffen; instrument; guitar), (Francis Lickerish; instrument; guitar), with isomorphism type (2)(1). The SPARQL query decomposes as intersection of two projections, originating from the seed entities. We can look at them separately.

- For seed node *Yehonatan Geffen*, the 2-hop path connecting it to the answer node is encoded by the query

  ```
  SELECT ?answer WHERE { wd:Q2911403 wdt:P40 ?child .
  ?child wdt:P1303 ?answer. }
  ```

  which returns three different answers: *guitar*, *piano*, *voice*.
- For seed node *Francis Lickerish*, the 1-hop path connecting it to the answer node is encoded by the query `SELECT ?answer WHERE { wd:Q3720616 wdt:P1303 ?answer. }`, which returns a single answer, *guitar*.

Therefore, this question contains redundant information, as it can be satisfyingly answered using just the seed entity *Francis Lickerish* (while using the other seed entity alone is not sufficient). The minimal answer subgraph is {(Francis Lickerish; instrument; guitar)}, with isomorphism type (1).

### A.4 EXAMPLES

We display, for exemplificative purposes, one datapoint from the train set of GTSQA, generated from Wikidata with the SynthKGQA framework.

```
Example

"id": 40513,

"question":  "Who directed the Italian film, originally in French,
that is based on 'The Vicomte of Bragelonne:  Ten Years Later'?",

"paraphrased_question":  "Who was the director of the Italian film,
originally in French, inspired by 'The Vicomte of Bragelonne:  Ten
Years Later'?",

"seed_entities":  ["Italy (Q38)", "French (Q150)", "The Vicomte of
Bragelonne:  Ten Years Later (Q769001)"],

"answer_node":  "Fernando Cerchio (Q503508)",
```

```
"answer_subgraph": [["Le Vicomte de Bragelonne (Q3228085)", "country
of origin (P495)", "Italy (Q38)"], ["Le Vicomte de Bragelonne
(Q3228085)", "original language of film or TV show (P364)", "French
(Q150)"], ["Le Vicomte de Bragelonne (Q3228085)", "based on (P144)",
"The Vicomte of Bragelonne: Ten Years Later (Q769001)"], ["Le
Vicomte de Bragelonne (Q3228085)", "director (P57)", "Fernando
Cerchio (Q503508)"]],

"sparql_query": "SELECT ?answer WHERE { ?film wdt:P495 wd:Q38;
wdt:P364 wd:Q150; wdt:P144 wd:Q769001; wdt:P57 ?answer.}",

"all_answers_wikidata": ["Q503508", "Q679016"],

"full_answer_subgraph_wikidata": [["Q2260875", "P495", "Q38"],
["Q2260875", "P364", "Q150"], ["Q2260875", "P144", "Q769001"],
["Q226087", "P57", "Q679016"], ["Q322808", "P495", "Q38"],
["Q3228085", "P364", "Q150"], ["Q3228085", "P144", "Q769001"],
["Q3228085", "P57", "Q503508"]],

"all_answers_wikikg2": ["Q503508"],

"full_answer_subgraph_wikikg2": [["Q3228085", "P364", "Q150"],
["Q3228085", "P57", "Q503508"], ["Q3228085", "P144", "Q769001"],
["Q3228085", "P495", "Q38"]],

"n_hops": 2,

"graph_isomorphism": "((1)(1)(1))",

"redundant": True,

"minimal_graph_isomorphism": "((1)(1))",

"minimal_seeds_and_queries": {"Q150-Q769001": "SELECT ?answer WHERE {
?a wdt:P364 wd:Q150. ?a wdt:P57 ?answer. ?a wdt:P144 wd:Q769001.}"}

"test_type": [],
```

Note that `answer_node` and `answer_subgraph` are, respectively, the answer node $a \in \mathcal{E}$ and ground-truth answer subgraph $\mathcal{G} \subset \mathcal{T}$ generated by the LLM together with the question. The `sparql_query` is then executed on Wikidata and WikiKG2 to retrieve all answers in the KGs (`all_answers_wikidata`; `all_answers_wikikg2`) and, after converting it to CONSTRUCT form, the full answer subgraphs realizing the query (`full_answer_subgraph_wikidata`; `full_answer_subgraph_wikikg2`). For the example above, we find that Wikidata (but not WikiKG2) contains one more acceptable answer (Henri Decoin, Q679016), due to the existence in the KG of a second movie satisfying all requirements in the question (Le Masque de fer, Q2260875); as a consequence, `full_answer_subgraph_wikidata` contains four more edges compared to $\mathcal{G}$, arising from these extra valid substitutions for the `?film` and `?answer` variables. Note that we only consider the answer graph $\mathcal{G}$ when computing the `graph_isomorphism`, as that encodes the logical steps required to reason over the question. However, the full answer subgraph should be used as target to evaluate the performance of KG retrievers.

`n_hops` measures the maximum distance (in $\mathcal{G}$) between a seed entity and the `answer_node`; it is determined in a unique way by `graph_isomorphism`. The question in the example contains redundant information (`redundant`); in the `minimal_seeds_and_queries` dictionary we provide the minimal set(s) $\mathcal{S}'$ of seed entities and the respective SPARQL queries $l_{q,\mathcal{S}'}$, as explained in Appendix A.3. The graph isomorphism of the minimal $\mathcal{G}_{\mathcal{S}'}$ can be found in `minimal_graph_isomorphism`. Finally, the attribute `test_type` is only used for questions in the test split of GTSQA, to classify their generalization type (in-distribution, unseen graph type, unseen relation type; see Section 4).

To judge the naturalness and reasonability of questions produced by the SynthKGQA framework, we present three randomly sampled questions for every graph isomorphism type in GTSQA with at least 50 questions in the training set. We observe that the paraphrasing operated by the LLM helps to make the questions sound more natural and human-like.

---

**Example questions and their paraphrased versions**

```
Graph isomorphism type (1)

  Question:  On which continent is Palmer Land located?
  Paraphrased Question:  Which continent is Palmer Land situated on?

  Question:  What was the military rank of Pierre Gaston-Mayer?
  Paraphrased Question:  What military rank did Pierre Gaston-Mayer
  hold?

  Question:  What was the noble title held by Sir Edward Kerrison,
  1st Baronet?
  Paraphrased Question:  What noble title did Sir Edward Kerrison,
  1st Baronet, hold?

Graph isomorphism type (2)

  Question:  Who is the head of government of the capital of the
  canton of Mérignac-1?
  Paraphrased Question:  Who is the leader of the government in the
  capital of the Mérignac-1 canton?

  Question:  Who is the producer of the series that the episode
  1930-talet is part of?
  Paraphrased Question:  Who produced the series that includes the
  episode titled "1930-talet"?

  Question:  Who composed the anthem of Tuvalu?
  Paraphrased Question:  Who is the composer of Tuvalu's national
  anthem?

Graph isomorphism type (3)

  Question:  In which country is the administrative entity in which
  Arhavi is located, located?
  Paraphrased Question:  Which country is home to the administrative
  region where Arhavi is situated?

  Question:  Where is the mother of the spouse of Ruprecht V of
  Nassau buried?
  Paraphrased Question:  Where is Ruprecht V of Nassau's
  mother-in-law buried?

  Question:  Who is the architect of the building occupied by the
  sports team that Raitis Grafs played for?
  Paraphrased Question:  Who designed the building where the sports
  team that Raitis Grafs played for is located?

Graph isomorphism type (1)(1)

  Question:  Which company founded by Henry Herbert Collier has its
  headquarters in Plumstead?
  Paraphrased Question:  Which company, established by Henry Herbert
  Collier, is headquartered in Plumstead?

  Question:  Which resident of District 2 was killed by Thresh?
  Paraphrased Question:  Who was the resident of District 2 that
  Thresh killed?

  Question:  Which film produced by UK Film Council was based on The
  Picture of Dorian Gray?
  Paraphrased Question:  What movie produced by the UK Film Council
  was inspired by The Picture of Dorian Gray?

Graph isomorphism type (1)(1)(1)

  Question:  Which singer, born in Suphan Buri, holds citizenship of
  Thailand and plays the guitar?
  Paraphrased Question:  Which Thai singer from Suphan Buri plays the
  guitar?
```

```
   Question:  Which film produced by Metro-Goldwyn-Mayer, whose main
   subject is baseball, was produced by Clarence Brown?
   Paraphrased Question:  What baseball-themed film made by
   Metro-Goldwyn-Mayer was directed by Clarence Brown?

   Question:  Who participated in the 2002 FIFA World Cup and played
   for both GNK Dinamo Zagreb and Sevilla FC?
   Paraphrased Question:  Which player took part in the 2002 FIFA
   World Cup and played for both GNK Dinamo Zagreb and Sevilla FC?

Graph isomorphism type (2)(1)

   Question:  Which team that plays association football is the
   occupant of the stadium operated by Fenerbahçe Sports Club?
   Paraphrased Question:  Which football team plays at the stadium
   managed by Fenerbahçe Sports Club?

   Question:  What is the city that is both the headquarters location
   of the developer of Star Wars:  Shadows of the Empire (video game)
   and the narrative location of Sudden Impact?
   Paraphrased Question:  Which city serves as the headquarters of the
   developer for the video game Star Wars:  Shadows of the Empire and
   is also the setting for Sudden Impact?

   Question:  Which street, used for utility cycling, serves as a
   terminus for a street that is itself a terminus at Frankfurter Tor?
   Paraphrased Question:  Which street designed for utility cycling
   acts as a terminus for another street that ends at Frankfurter Tor?

Graph isomorphism type ((1)(1)(1))

   Question:  Which cast member acted in a Spanish-language film that
   was filmed in Mexico and belongs to the exploitation film genre?
   Paraphrased Question:  Which cast member starred in a
   Spanish-language film shot in Mexico that falls under the
   exploitation genre?

   Question:  Who is the head of government of the municipality that
   is contained within Valdizarbe and shares borders with both Uterga
   and Enériz?
   Paraphrased Question:  Who is the head of government for the
   municipality in Valdizarbe that borders both Uterga and Enériz?

   Question:  Which award was received by the English-language film
   directed by Terence Davies and starring Freda Dowie?
   Paraphrased Question:  What award did the English-language film
   directed by Terence Davies and featuring Freda Dowie win?

Graph isomorphism type (2)(1)(1)

   Question:  Which goalkeeper who participated in the 1958 FIFA World
   Cup died in a town located in the Eastern Time Zone?
   Paraphrased Question:  Which goalkeeper from the 1958 FIFA World
   Cup passed away in a town in the Eastern Time Zone?

   Question:  Which businessperson who was a cast member of a reality
   television show also attended the University of Pennsylvania?
   Paraphrased Question:  Which reality TV star who is also a
   businessperson went to the University of Pennsylvania?

   Question:  Which person died of pneumonia in Vienna and has an
   asteroid belt minor planet named after them?
   Paraphrased Question:  Who passed away from pneumonia in Vienna and
   has a minor planet in the asteroid belt named in their honor?

Graph isomorphism type (1)(1)(1)(1)(1)

   Question:  Which musical film directed by Challis Sanderson
   features both Robb Wilton and Kitty McShane as cast members and
   has Desmond Dickinson as its director of photography?
```

```
    Paraphrased Question:  What musical film directed by Challis
    Sanderson stars Robb Wilton and Kitty McShane, with Desmond
    Dickinson as the director of photography?

    Question:  Which video game is both an action and platform game,
    distributed on Nintendo game card for Nintendo DS and published in
    North America?
    Paraphrased Question:  What action and platform game is available
    on a Nintendo game card for the Nintendo DS and was published in
    North America?

    Question:  Which minor planet in the asteroid belt was discovered
    at Osservatorio Astronomico Sormano by both Francesco Manca and
    Piero Sicoli and is directly preceded by 9110 Choukai?
    Paraphrased Question:  What is the name of the minor planet in the
    asteroid belt, discovered by Francesco Manca and Piero Sicoli at
    the Osservatorio Astronomico Sormano, that comes right before 9110
    Choukai?
```

# B    STATISTICS OF GTSQA

We use ogbl-wikikg2 (Hu et al., 2020) as the base KG to construct GTSQA. This is a KG extracted from a 2015 Wikidata dump, containing a curated set of 2.5M nodes and 535 relation types, that we find are good candidates for the construction of natural-sounding questions. As part of the validation and filtering pipeline, by leveraging the generated SPARQL queries, we reject datapoints where any of the edges in the ground-truth answer subgraph inside ogbl-wikikg2 encode stale facts, i.e., edges that are not contained in the most up to date version of Wikidata[3]. In the future, GTSQA can be easily kept up to date by repeating the filtering process against new Wikidata dumps, or by replacing stale ground-truth subgraphs with the current ones, as retrieved from Wikidata via the provided SPARQL queries.

An overall view of the statistics of GTSQA is provided in Table 1. The questions in the dataset involve 68,520 unique entities, drawn from ogbl-wikikg2. The distribution of relation types is shown in more detail in Figure B1: out of the 368 unique relation types used in the dataset, only the top-200 (when sorting by overall frequency) appear in the train set, while the remaining ones are reserved for testing. Figure B2 shows the distribution of the number of seed entities, hops (in the ground-truth answer subgraph) and answers. Questions in the test set are significantly harder, requiring to perform more hops, or to combine reasoning chains from multiple seed entities. Our data-filtering pipeline focuses on selecting highly-factual and non-ambiguous questions; as a consequence, 73.9% and 85.9% of questions in the train and test set, respectively, have a single answer in Wikidata, and only a negligible fraction have more than five.

We also compare the size of the ground-truth answer subgraph and the full answer subgraph retrieved from Wikidata by submitting the SPARQL query in CONSTRUCT form. As recalled in Appendix A.4, the full subgraph may contain additional edges, originating from the presence of multiple answers and/or multiple choices for the intermediate entities not specified in the query. In practice, as shown in Figure B3, for GTSQA this excess in the number of edges remains always limited (only in 28.4% and 15.5% of train and test questions, respectively, the two graphs do not coincide). Finally, we report on statistics on redundant information in the train set of GTSQA (as stressed in Section 4, no redundancy is present in test questions). We find that 25.76% of training questions contain some degree of redundancy; Figure B4 shows the distributions of graph isomorphism types for the ground-truth answer subgraph $\mathcal{G}$ and the minimal subgraph $\mathcal{G}_{\mathcal{S}'}$ (see Appendix A.3).

---

[3]At the time of dataset construction: `https://dumps.wikimedia.org/wikidatawiki/20250720/`

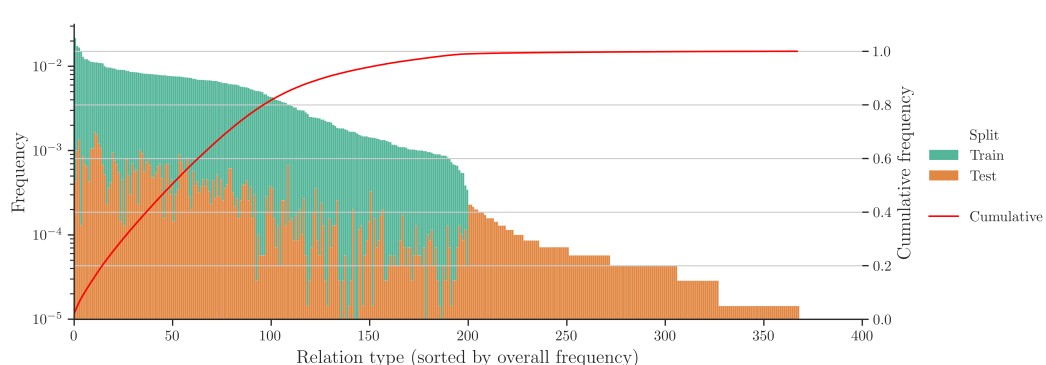

Figure B1: Frequency of relation types of edges in the ground-truth answer subgraphs of questions in GTSQA. The 168 least-occurring relation types (tail of the distribution) are reserved for questions in the test set, to test zero-shot generalization abilities of KG retriever models.

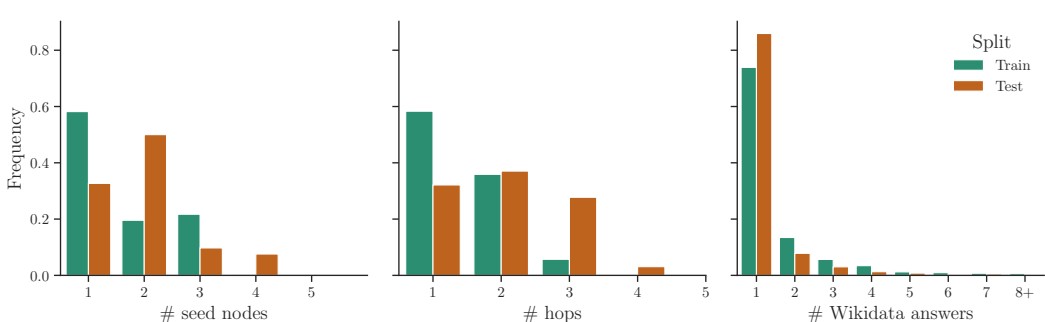

Figure B2: Statistics on the number of seed entities (*left*), the number of answer entities in Wikidata (*right*), and the maximum number of hops from seed to answer nodes along ground-truth paths (*center*), for questions in the train and test set of GTSQA.

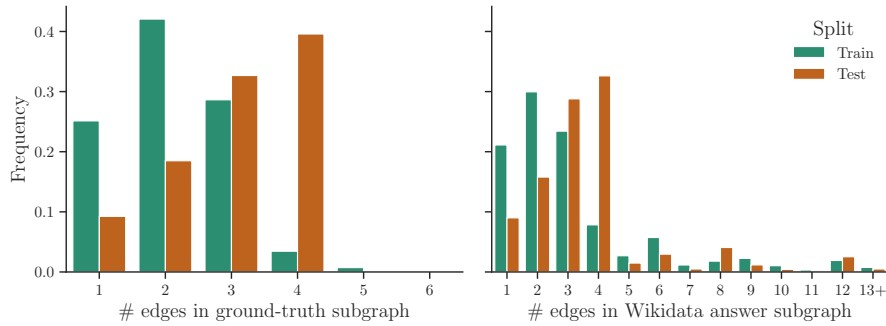

Figure B3: Statistics on the number of edges in the ground-truth answer subgraph (*left*) and the full answer subgraph in Wikidata (*right*), for questions in the train and test set of GTSQA. The full answer subgraph can contain more edges than the ground-truth subgraph, if the question has multiple answers, or if any of the intermediate nodes is not uniquely determined.

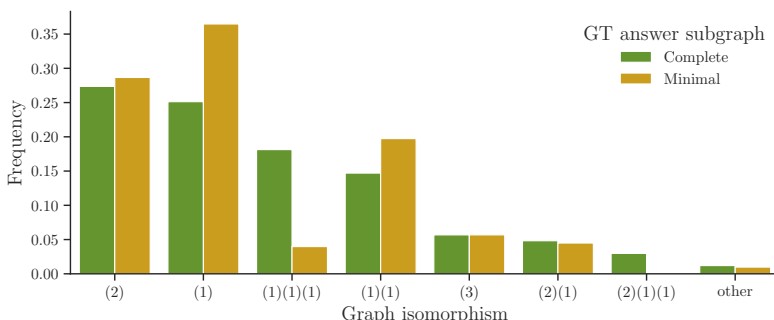

Figure B4: Frequency of main graph isomorphism types in the train set of GTSQA, distinguishing between the complete ground-truth answer subgraph $\mathcal{G}$ and the minimal subgraph $\mathcal{G}_{\mathcal{S}'}$, obtained from the complete one after discarding redundant information.

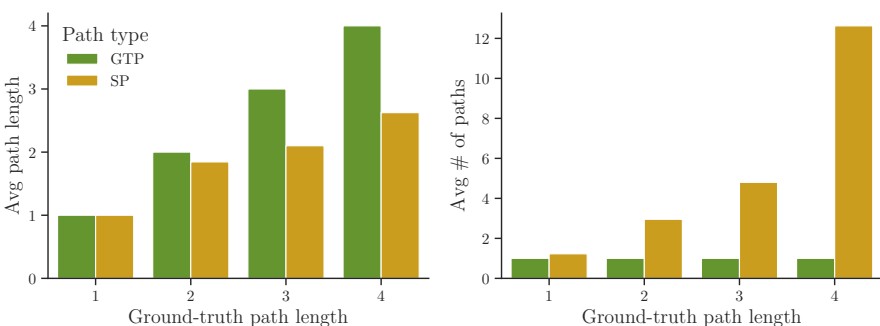

Figure B5: Comparison of ground-truth paths (GTP) and shortest paths (SP) connecting a question seed node to the answer node, in the train split of GTSQA. *Left:* for questions where the ground-truth paths require multiple hops, the distance from the seed node to the answer node along the shortest path increases sub-linearly (shortcuts). *Right:* as the distance between seed and answer node increases, the number of parallel paths of minimal length between them grows exponentially.

## C ADDITIONAL DETAILS ON EXPERIMENTS

### C.1 CONSTRUCTION OF QUESTION-SPECIFIC GRAPHS

An often overlooked fact in KGQA benchmarks is that many state-of-the-art KG retrievers, especially those that need to perform breadth/depth-first exploration of the KG from the seed entities (e.g., Luo et al. (2024), Luo et al. (2025)) or that include Graph Neural Networks in their architectures (e.g., Mavromatis & Karypis (2025), Li et al. (2025)), are too expensive to run on KGs that have more than $\sim 10^5$ edges. This limitation, in practice, is tackled by performing the retrieval on a smaller subgraph of the KG, which is independently sampled for each question, e.g., by taking the full $k$-hop neighborhood of the seed entities in the KG, and then pruning it down to a few tens of thousands of

Table B2: Overlap of triples in ground-truth (GT) answer subgraph and triples on shortest paths (SP) between seed nodes and answer nodes. Questions are grouped by the maximum number of hops in the ground-truth subgraph (# hops); the other columns report the average metric values.

| # hops | % GT triples in SP | % SP triples in GT | # GT triples | # SP triples |
|--------|--------------------|--------------------|--------------|--------------|
| 1 | 100.0 | 91.0 | 2.37 | 2.67 |
| 2 | 89.8 | 52.7 | 3.08 | 19.6 |
| 3 | 54.7 | 29.6 | 3.6 | 46.3 |
| 4 | 27.5 | 13.3 | 4 | 100.4 |

edges with algorithms like Personalized PageRank (Page et al., 1998). This was the case for Luo et al. (2024), where such question-specific graphs were constructed (with $k = 2$) from Freebase (Bollacker et al., 2008) for the questions in WebQSP (Yih et al., 2016) and CWQ (Talmor & Berant, 2018). These graphs were then used by many others in following papers for benchmarking new models on these two widely-used datasets. They are not, however, part of the official datasets, hence there is no guarantee on their adoption. It is important to note, in fact, that the selection of these starting graphs can strongly impact the final performance statistics, potentially over-representing (if they are too easy/small) or under-representing (if they are not checked to still contain ground-truth paths) the retriever's capabilities. For this reason, retrievers that are benchmarked on different sets of questions-specific graphs should not be directly compared (even though the test questions, and the underlying full KG, are the same), and care should be taken when comparing them with models that instead are able to perform retrieval from the full KG (e.g., Sun et al. (2024)). However, this crucial detail on experimental setup is often not reported in papers, making comparisons unreliable.

To address this problem, together with GTSQA we release an official set of question-specific graphs (each containing up to 30,000 edges) for all questions in the train and test set. These are the only graphs that should be used by anyone wishing to train or benchmark KG-RAG models on GTSQA when retrieving from the full KG is not possible, to ensure fair comparisons. They are extracted from ogbl-wikikg2 with a similar approach to Luo et al. (2024), starting from the full undirected 3-hop (4-hop for questions requiring 4 hops) neighborhood of seed entities, and then pruning it down to the edges connecting the nodes with the top-2500 scores as assigned by Personalized PageRank (with personalization values concentrated in the seed nodes). If any of the edges in the (full) ground-truth answer subgraph of the question have been dropped as a result of pruning, they are re-added to the graph to guarantee that perfect retrieval is still possible. However, we observe empirically that this final step can unfairly bias retrievers towards the ground-truth edges that have been re-added. To ensure a challenging task, we also add to the graph (as confounders) all edges along paths originating from the seed nodes, of the same metapath (sequence of relation types) as the corresponding ground-truth paths that lead to the answer node.

## C.2 SPECIFICATIONS OF EVALUATED MODELS

We provide details and specifications on the KG-RAG models included in our benchmarks. For all of them, we follow original implementations as close as possible.

**Think-on-Graph (Sun et al., 2024), Plan-on-Graph (Chen et al., 2024)** We modified the original codebases (with the PoG one being based on the one from ToG) to perform retrieval from the ogbl-wikikg2 KG. The search and prune steps of the KG exploration algorithm use a width of 5 and a maximum depth of 4 to enable the retrieval of all paths in the ground-truth answer subgraphs of multi-seed, multi-hop questions in the test set of GTSQA. Note that ToG, by its original implementation, reverts to answering using only the LLM knowledge if the maximum search depth is reached without the model being confident it has retrieved enough information; in this case, we treat the retrieved subgraph as being empty. The LLM performing the graph exploration (and task decomposition and memory updating for PoG) is the same used for the final reasoning, namely GPT-4o-mini.

**SR (Zhang et al., 2022)** As in the original implementation, we use RoBERTa$_{BASE}$ (Liu et al., 2019) to predict the next relation type $r_N \in \mathcal{R} \cup \{END\}$ in a path $(r_1, \ldots, r_{N-1})$ from seed to answer node, conditioning on the question $q$. This is performed by fine-tuning the text-encoder to align the embeddings of $[q; r_1; \ldots; r_{N-1}]$ and $r_N$, with a contrastive approach that uses positive and negative pairs constructed from ground-truth paths (from seed to answer) in the train set. At inference time, paths are predicted by conducting a beam search based on possible continuation candidates; we impose a maximum path length of 4 and set the number of beams to 5. The subgraph is retrieved by looking for all possible (undirected) realizations of the predicted relation paths in the KG starting from the seed entities. Note that, while the original implementation used a Neural State Machine (He et al., 2021) to perform the final reasoning on the retrieved subgraph, we instead use an LLM, to align the last step of the pipeline with the other KG-RAG models evaluated in the paper.

**Reasoning on Graphs (Luo et al., 2024)** As in the original paper, we fine-tune LLama-2-Chat-7B (Touvron et al., 2023) to auto-regressively predict the relation paths $(r_1, \ldots, r_N)$ (and specifying the direction of each edge), originating from the seed nodes, that should be useful to answer a question

$q$. We use all ground-truth paths for the questions in the train set as fine-tuning data. At inference time we ask the LLM to propose relation paths using beam search (5 beams) which are then used to retrieve the subgraph via breadth-first search. We adopt the plug-and-play version of RoG, which allows us to use a different LLM (GPT-4o-mini) to perform the final reasoning on the retrieved triples.

**Graph-Constrained Reasoning (Luo et al., 2025)**   While RoG can hallucinate non-existing relation paths, the follow-up work GCR constraints the path decoding to actual paths in the KG. However, this comes at significant costs in terms of overhead, as it requires to first index in a KG-Trie all paths (up to a fixed length) originating from the seed entities, retrieved via depth-first transversal of the graph. We find that, even when working with the smaller question-specific graphs from Appendix C.1, building such index for paths of length $> 2$ requires an unpractical amount of time, which strongly limits the applicability of the method to real-world scenarios where the index needs to be built on-the-fly (as it happens for questions that have not been seen before). For this reason, we test GCR with a maximum path length of 2 (which is also the default for experiments in Luo et al. (2025)), despite being aware that a significant fraction of questions in GTSQA require reasoning over longer paths. While we still include GCR in the benchmark of GTSQA, these limitations lead us to exclude GCR from the case-study in Section 6. As in the original implementation, we use LLama-3.1-8B-Instruct (AI@Meta, 2024) as LLM to generate paths, and fine-tune it with the same data used for RoG. At inference time, we generate 10 explicit paths form the seed entities through graph-constraint decoding, and then discard duplicated paths to obtain the final retrieved subgraph.

**SubgraphRAG (Li et al., 2025)**   SubgraphRAG assigns relevance scores to all edges in the question-specific graphs (Appendix C.1), by combining text embeddings with message passing. In particular, $p((h, r, t)|q) \propto \text{MLP}([z_q; z_h; z_r; z_t; z_\tau])$, where $z_q, z_h, z_r, z_t$ are text embeddings from gte-large-en-v1.5 (Li et al., 2023) for the question $q$ and the labels of $h, r, t$. The embedding $z_\tau$ is constructed from the GNN embeddings of the $h$ and $t$ nodes, after 2 layers of message passing starting from the one-hot representation of nodes in the graph provided by the labeling trick ($\mathbf{1}$ if the node is a seed entity, $\mathbf{0}$ otherwise; Zhang et al. (2021)). We train GNN and MLP with cross-entropy loss to assign high scores to the triples in the ground-truth answer subgraphs, on the train split of GTSQA. At inference time, we retrieve the subgraph made of the triples with top-200 scores, similarly to the experiments in Li et al. (2025) (after checking that performance starts to plateau when increasing the subgraph size further, Figure C7).

### C.3   Additional Results

Figure C8 presents a comparison of the recall of ground-truth triples for the evaluated KG-RAG models on GTSQA, complementing the data in Figure 2. For trainable models, we also show the results disaggregated by generalization type of the test question in Figure C9.

In Figure C10 we provide additional results for the analysis in Section 6, quantifying improvements across different metrics for KG-RAG models trained on the ground-truth subgraphs in GTSQA, compared to training on shortest paths between seed and answer nodes, as conventionally done.

#### C.3.1   GPT-5-MINI AS REASONING MODEL

We repeat the main analysis of Section 5, but using the newer GPT-5-mini (OpenAI, 2025) as the model performing the final reasoning on the KG-retrieved subgraphs, to understand how the choice of the LLM influences Hits (EM). While all models score significantly higher than with GPT-4o-mini (Table C3), the relative ranking remains the same as observed before, and no different patterns or behaviors arise when looking at individual graph isomorphism types (Figure C11) or at different generalization abilities required to answer (Figure C12).

We notice that SubgraphRAG is the model benefiting the most from using a more recent LLM to reason over the retrieved subgraph. This appears to be due to an improved utilization of long context, filtering out the noise in the augmented prompt when presented with a large number of retrieved triples. Indeed, while with GPT-4o-mini we observed a gap between model response quality as measured by Hits (EM) and recall of ground-truth triples when the number of retrieved triples was increased, GPT-5-mini exhibits continued improvements in Hits (EM) up to 500 triples, matching closely the steady increase in GT triple recall (Figure C7). This is reflected, across all models, in

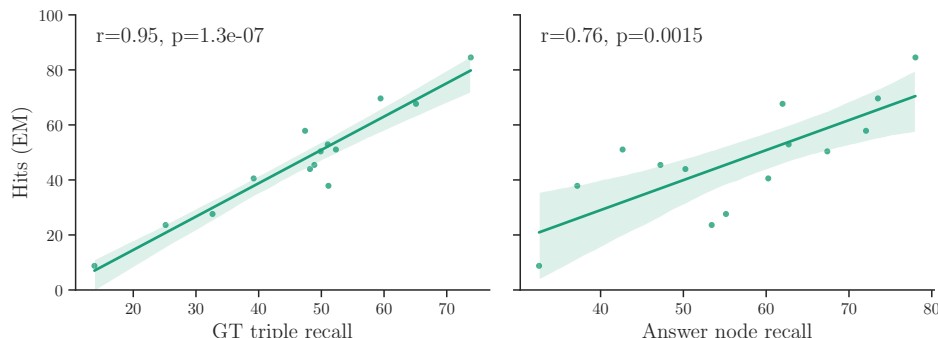

Figure C6: Correlation between Hits (EM) and recall of ground-truth triples (*left*) and of answer nodes (*right*), for questions in the test set of GTSQA. Each dot represents the average performance of the evaluated KG-RAG models on a different graph isomorphism type. While both recall variables have a positive linear correlation with predictive performance, recall of ground-truth triples is a significantly stronger predictor.

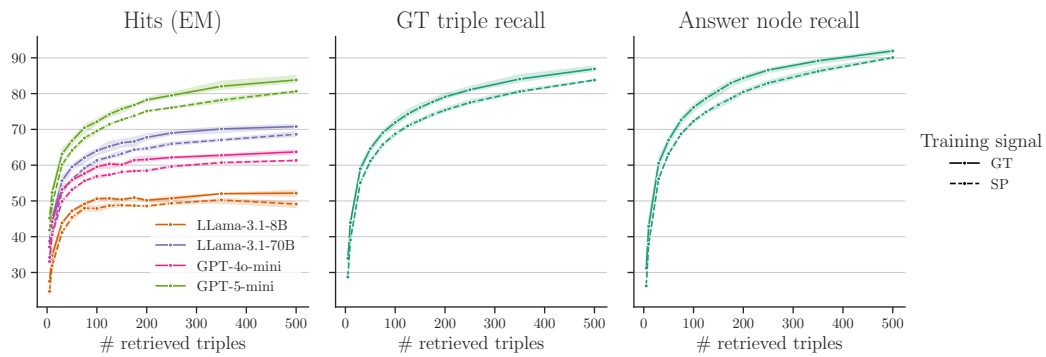

Figure C7: Performance of SubgraphRAG (with different LLMs doing the final reasoning) on the test set of GTSQA, as we increase the size of the retrieved subgraph. More capable LLMs are less sensitive on the noise in the retrieved data, with Hits (EM) tracking more closely the increase in recall of ground-truth triples and answer nodes. Across all LLM and subgraph sizes, training on the ground-truth answer subgraphs (GT) outperforms training on shortest path triples (SP).

an even stronger linear correlation between Hits (EM) and the recall of ground-truth triples, with Pearson correlation coefficient increasing from $r = 0.95$ ($p = 1.3\mathrm{e}{-7}$) for GPT-4o-mini (Figure C6) to $r = 0.97$ ($p = 1.5\mathrm{e}{-8}$) for GPT-5-mini. Training on GT triples (rather than shortest paths) proves to be even more beneficial to the final predictive accuracy than we observed for GPT-4o-mini (Figure C7).

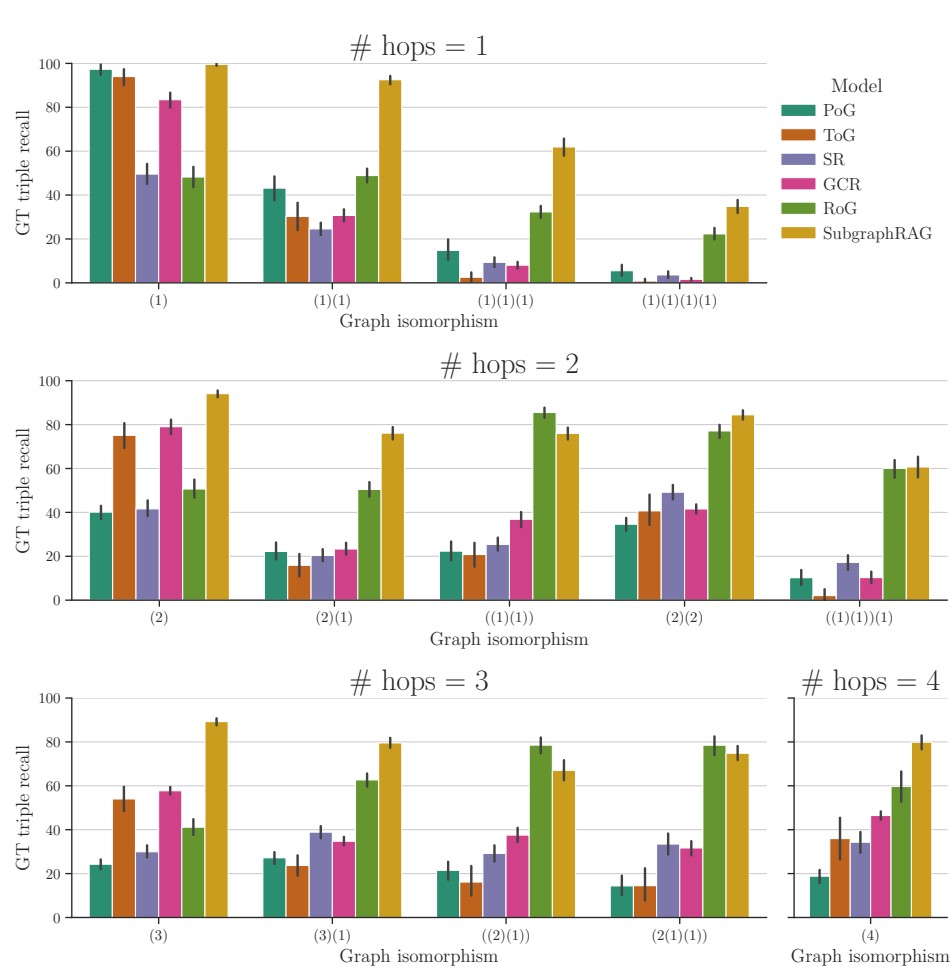

Figure C8: Recall of triples in ground-truth answer subgraph, for KG-RAG models on different graph isomorphism types.

Table C3: Benchmark of KG-RAG models on GTSQA with GPT-5-mini as final reasoning model.

| Category | Model | EM | | Ground-truth triples | | | Answer nodes | | |
|---|---|---|---|---|---|---|---|---|---|
| | | Hits | Recall | Recall | Precision | F1 | Hits | Recall | # triples |
| KG agent | PoG | 41.74 | 40.32 | 36.07 | 37.14 | 34.07 | 35.76 | 33.56 | 3.72 |
| | ToG | 53.02 | 51.19 | 32.78 | 7.00 | 10.80 | 36.13 | 35.06 | 8.00 |
| Path-based | SR | 51.75 | 49.75 | 30.22 | 3.44 | 5.69 | 50.25 | 49.39 | 72.94 |
| | GCR | 56.58 | 54.59 | 40.71 | 27.21 | 29.83 | 47.10 | 45.53 | 6.54 |
| | RoG | 67.47 | 65.09 | 54.69 | 24.04 | 27.00 | 72.91 | 71.84 | 72.15 |
| All-at-once | SubgraphRAG (200) | 78.24 | 74.28 | 79.09 | 1.29 | 2.53 | 85.33 | 84.36 | 199.61 |

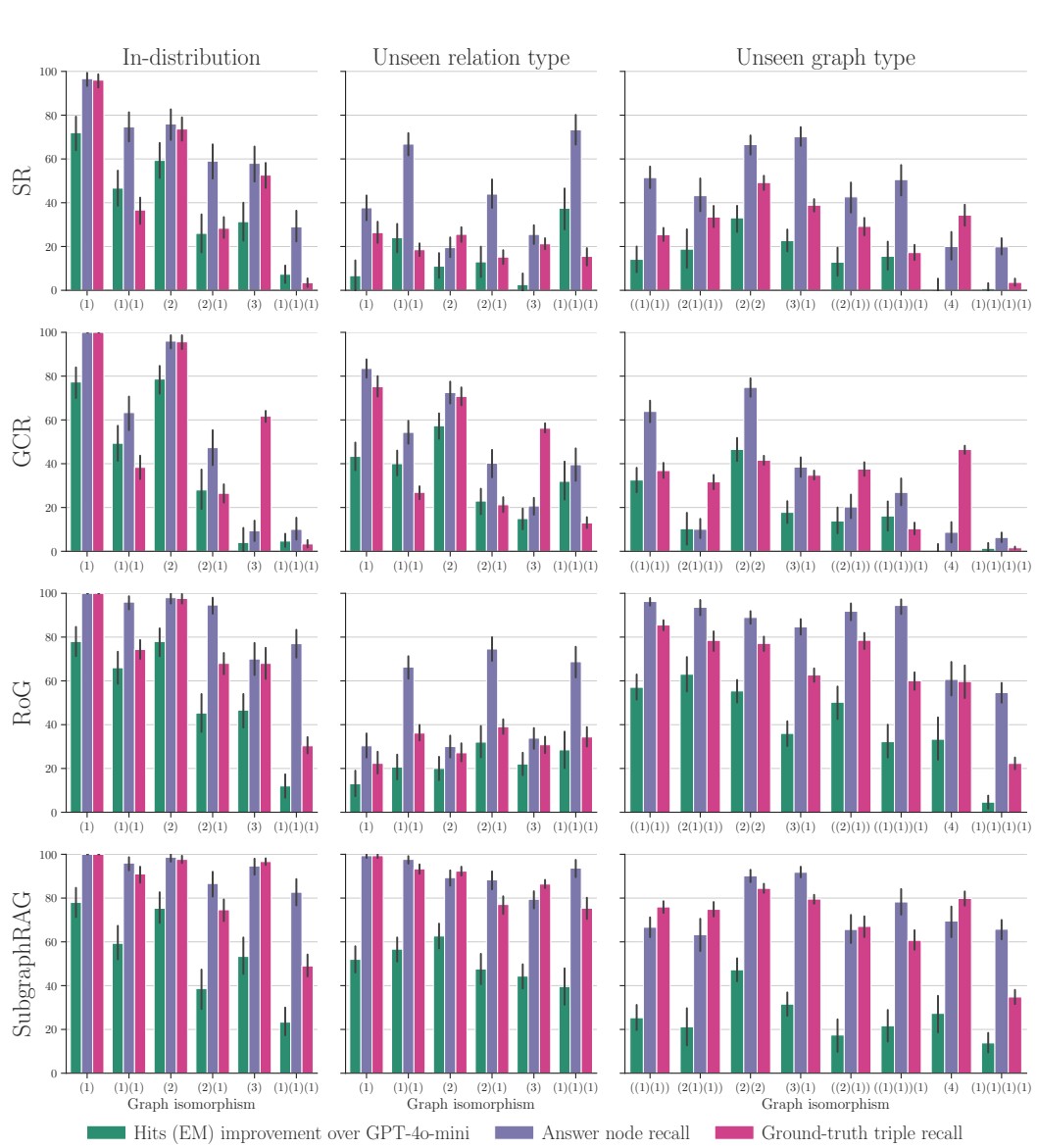

Figure C9: Detailed analysis of generalization abilities of trainable KG-RAG models, on different graph isomorphism types. We measure EM performance in terms of difference with the EM of the baseline (GPT-4o-mini, no RAG).

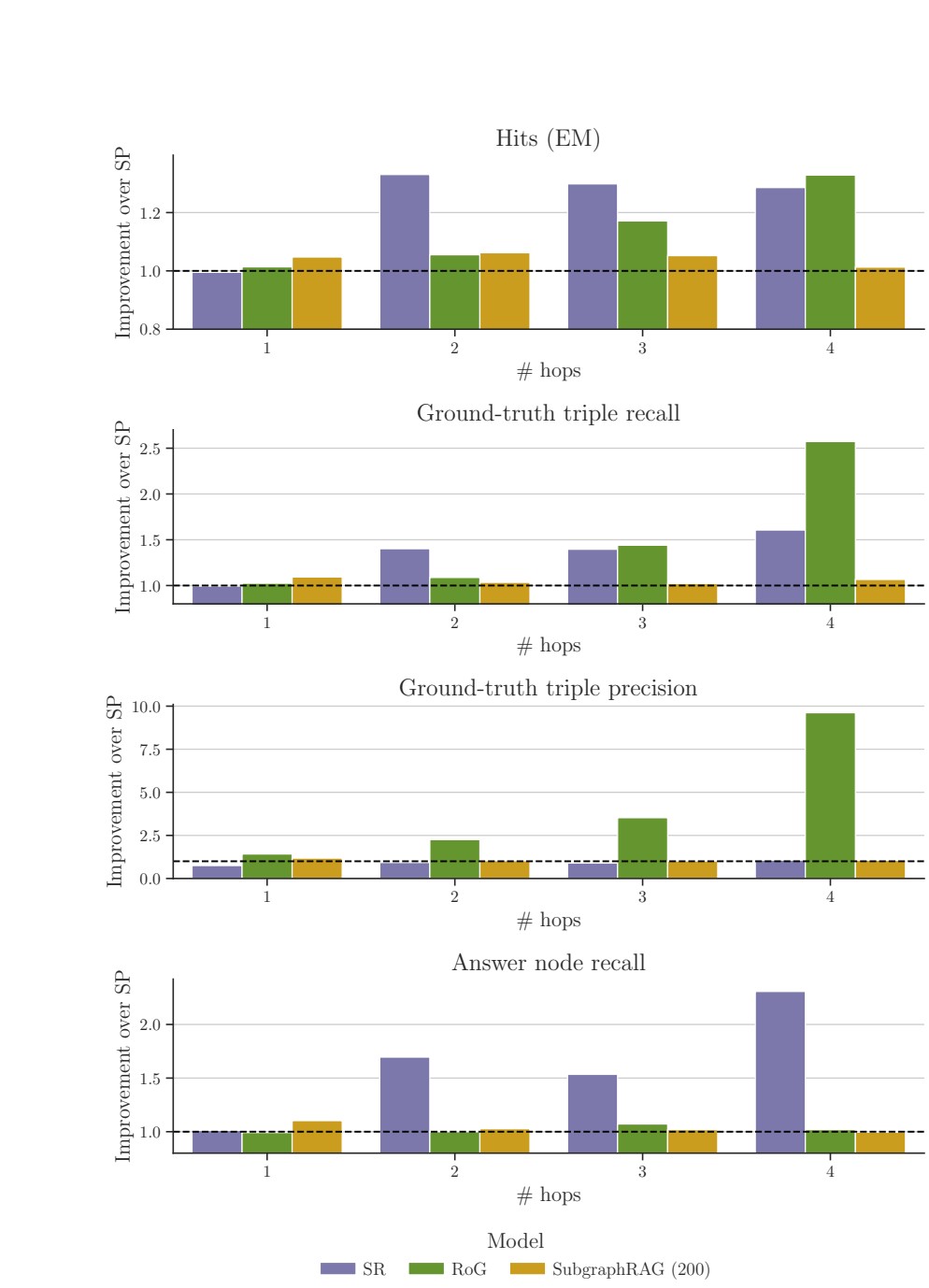

Figure C10: Comparison of predictive statistics when training models on ground-truth (GT) answer subgraphs compared to training on shortest paths (SP), for questions requiring different number of hops. The y-axis measures the GT/SP ratio of the averages of the statistic on the test set (over three distinct training runs); if the ratio is above the dashed line, training on the ground-truth subgraph is better.

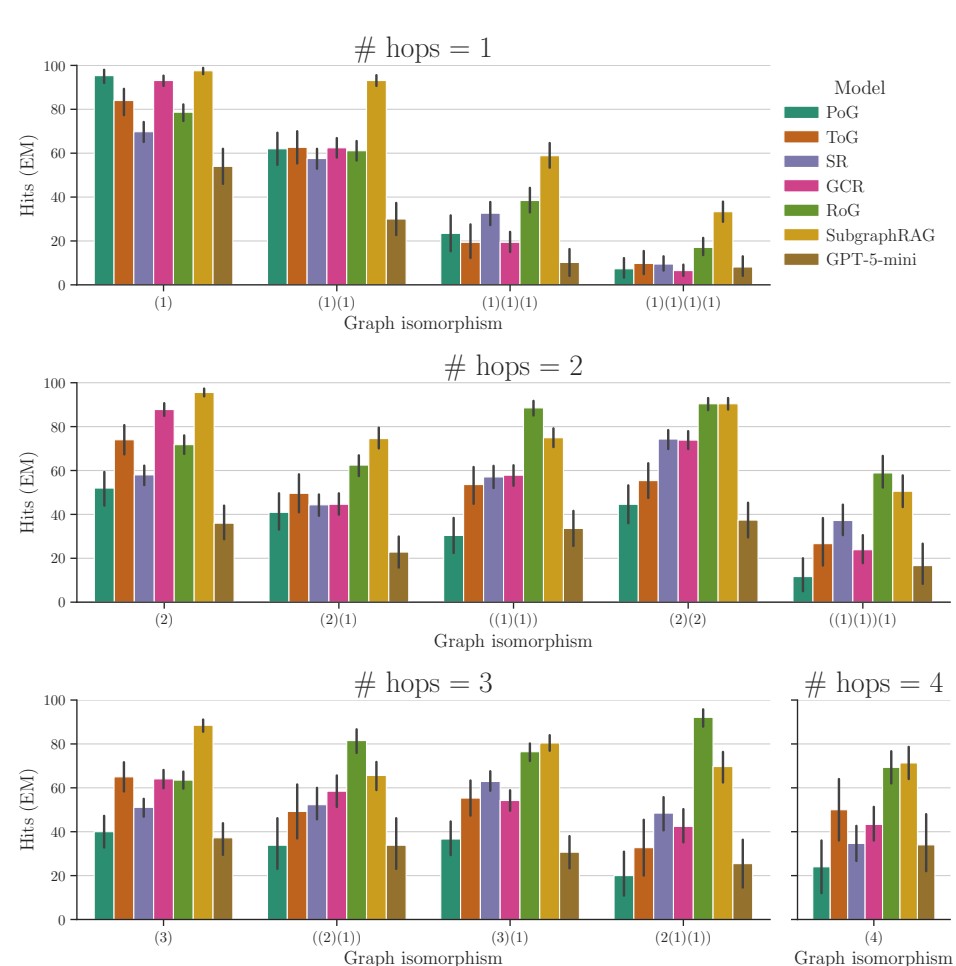

Figure C11: Hits (EM) of KG-RAG models on different graph isomorphism types, compared to the baseline (GPT-5-mini, no RAG).

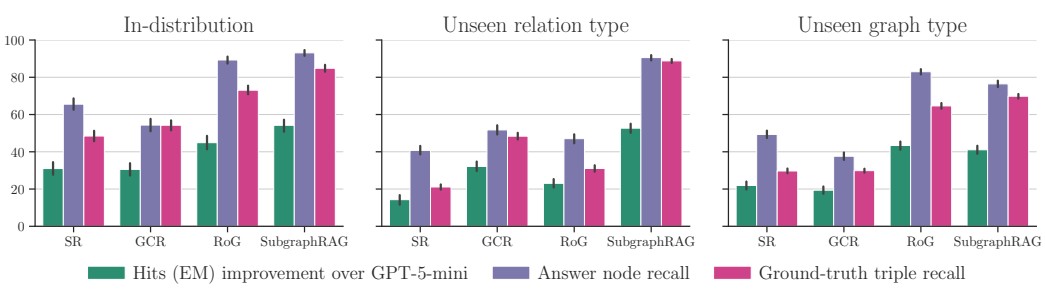

Figure C12: Generalization abilities of trainable KG-RAG models. We measure EM performance in terms of difference with the EM of the baseline (GPT-5-mini, no RAG).

## D   CASE STUDY OF FAILURE CASES

By presenting explicit examples, we discuss some common failure cases of different KG-RAG models, identified through the analysis in Section 5 and manual exploration of the experimental data. For all examples, the provided statistics are averaged over three runs of the models.

**Multi-seed questions**   As remarked in Section 5, all evaluated models perform poorly when the question requires expanding and combining reasoning paths from 3 or more seed entities. RoG and SubgraphRAG (and, to a lesser extent, SR) show better performance than other methods, as they typically retrieve larger sets of triples, which often mitigates the issue.

*Question*: What common genre do Johan Georg Schwartze and the painting by Alexander Andreyevich Ivanov in the Russian Museum share?

*Seed entities*: Johan Georg Schwartze, Alexander Andreyevich Ivanov, Russian Museum.

*Answers*: landscape art.

*Ground-truth answer subgraph*: (Johan Georg Schwartze; genre; landscape art), (Water and Stones near Palazzuola; creator; Alexander Andreyevich Ivanov), (Water and Stones near Palazzuola; collection; Russian Museum), (Water and Stones near Palazzuola; genre; landscape art).

*Graph isomorphism*: ((1)(1))(1).

*Generalization type*: unseen graph type.

| Model | Hits (EM) | Ground-truth triples | | | Answer nodes | |
|---|---|---|---|---|---|---|
| | | Recall | Precision | F1 | Recall | # triples |
| GPT-4o-mini | 0.00 | - | - | - | - | - |
| PoG | 0.00 | 25.00 | 14.29 | 18.18 | 100.00 | 7.00 |
| ToG | 0.00 | 0.00 | 0.00 | 0.00 | 0.00 | 0.00 |
| SR | 0.00 | 33.33 | 1.50 | 2.82 | 100.00 | 107.67 |
| RoG | 100.00 | 100.00 | 5.80 | 10.96 | 100.00 | 69.00 |
| GCR | 0.00 | 25.00 | 41.67 | 30.56 | 100.00 | 2.67 |
| SubgraphRAG | 0.00 | 75.00 | 1.50 | 2.94 | 100.00 | 200.00 |

- **GPT-4o-mini**: without any augmentations, the answer provided by the LLM is "Johan Georg Schwartze and the painting by Alexander Andreyevich Ivanov in the Russian Museum both share the genre of historical painting. The answer is {historical painting}". No information on the painting is provided, making hard to assess where the hallucination originates from.

- **PoG**: the model is able to retrieve the ground-truth triple (Johan Georg Schwartze; genre; landscape art). It also expands the search from the other two seed entities, retrieving multiple 1-hop paths from each of them, but it is unable to identify the painting that satisfies at the same time both the conditions in the question.

- **ToG**: the model proceeds to iteratively expand the search from only one of the seed nodes (Johan Georg Schwartze). When the maximum depth of exploration is reached, since not enough information to answer the question has been collected, ToG reverts to only answering using the LLM knowledge (Appendix C.2).

- **SR**: the unique relation paths predicted by the model are: (location, genre); (location, genre, field of work); (genre, depicts); (located in the administrative territorial entity). When looking for (possibly partial) realizations of these relation paths in the KG (in any direction), starting from the seed entities, the model retrieves the ground-truth triple (Johan Georg Schwartze; genre; landscape art). It also retrieves (Water and Stones near Palazzuola; location; Russian Museum), (Water and Stones near Palazzuola; genre; landscape art), through the metapath (location; genre) starting from the seed node "Russian Museum" (a parallel path, still valid, compared to the one in the ground-truth answer subgraph). However, no relation types pertaining to painting authorship are predicted.

- **RoG**: the model predicts much better relation paths than SR, namely: (genre); (inverse of: collection, genre); (inverse of: creator, genre); (collection, genre). As a result, all ground-truth triples are correctly retrieved (together with ∼ 65 more).

- **GCR**: as we observe frequently in results from this method, GCR focuses on a single seed entity when decoding graph-constrained paths. Here, after de-duplication of the outputs, we find ourself with only two paths: *Johan Georg Schwartze → genre → landscape art*; *Johan Georg Schwartze → genre → portrait*. Thus, only one triple in the ground-truth answer subgraph is retrieved, namely (Johan Georg Schwartze; genre; landscape art).

- **SubgraphRAG**: the top-200 triples retrieved by the model contain three ground-truth edges, namely: (Johan Georg Schwartze; genre; landscape art), (Water and Stones near Palazzuola; creator; Alexander Andreyevich Ivanov), (Water and Stones near Palazzuola; collection; Russian Museum). The model is therefore able to identify the painting satisfying the two conditions in the query, but not to make the additional hop from it to the answer node. As highlighted in Section 5, this is likely due to the poor generalization abilities of SubgraphRAG to new graph isomorphisms, as no questions requiring additional projections after the intersection of paths from different seed entities have been observed during training.

**Multi-hop questions**   Questions requiring more than two hops, even in the presence of a single seed entity, pose significant challenges to most KG retrievers.

*Question*: Who is the lyricist of the national anthem for the country that is home to the Nauru Reed Warbler?

*Seed entities*: Nauru Reed Warbler.

*Answers*: Margaret Hendrie.

*Ground-truth answer subgraph*: (Nauru Reed Warbler; endemic to; Nauru), (Nauru; anthem; Nauru Bwiema), (Nauru Bwiema; lyrics by; Margaret Hendrie).

*Graph isomorphism*: (3).

*Generalization type*: unseen relation type (relation "endemic to" not included in the train set).

| Model | Hits (EM) | Ground-truth triples | | | Answer nodes | |
| | | Recall | Precision | F1 | Recall | # triples |
|---|---|---|---|---|---|---|
| GPT-4o-mini | 0.00 | - | - | - | - | - |
| PoG | 0.00 | 33.33 | 100.00 | 50.00 | 0.00 | 1.00 |
| ToG | 100.00 | 100.00 | 10.71 | 19.35 | 100.00 | 28.00 |
| SR | 66.67 | 66.67 | 10.25 | 17.67 | 66.67 | 33.00 |
| RoG | 0.00 | 0.00 | 0.00 | 0.00 | 0.00 | 0.00 |
| GCR | 0.00 | 66.67 | 40.00 | 50.00 | 0.00 | 5.00 |
| SubgraphRAG (200) | 100.00 | 100.00 | 1.50 | 2.96 | 100.00 | 200.00 |

- **GPT-4o-mini**: the answer provided by the LLM is "The Nauru Reed Warbler is found in Nauru. The national anthem of Nauru is 'Nauru Bwiema,' and the lyricist is the former president of Nauru, Hammer DeRoburt. The answer is {Hammer DeRoburt}". The model is therefore able to correctly identify the nation and the anthem, but it hallucinates the identity of the lyricist.

- **PoG**: the model correctly retrieves the first ground-truth triple (Nauru Reed Warbler; endemic to; Nauru). However, the agent then decides to prematurely stop the search and directly generate the answer (this is a common behavior of PoG on multi-hop questions, as observed in Section 5).

- **ToG**: in a scenario with a single seed entity, the KG retriever is able to effectively expand the search and fetch a set of $\sim 30$ triples containing all the ground-truth ones.

- **SR**: there are only a few relation types generating from the seed entity, hence the model is able to identify sensible relation paths. In two tries out of three, it predicts: (IUCN conservation status, IUCN conservation status, depicts); (endemic to, anthem); (endemic to, anthem, lyrics by); (endemic to, anthem, composer). This leads to retrieving all the ground-truth triples.

- **RoG**: the relation paths predicted by RoG are (country of origin, anthem, lyrics by); (country of citizenship, anthem, lyrics by); (inverse of: has part, lyrics by); (inverse of: native bird,

anthem, lyrics by). While the two ground-truth relation types seen during training ("anthem", "lyrics by") are consistently predicted, the model is unable to come up with good suggestions for the unseen relation type "endemic to", proposing instead known relation types with a similar meaning (e.g., "country of origin") or hallucinating inexistent relation types, such as "native bird". None of the proposed relation paths is realized in the KG starting from the seed entity, hence the set of retrieved triples is empty.

- **GCR**: the graph-constrained decoding from the seed entity leads to identifying the following paths: *Nauru Reed Warbler → taxon rank → species → inverse of: taxon rank → Nauru Reed Warbler*; *Nauru Reed Warbler → endemic to → Nauru → anthem → Nauru Bwiema*; *Nauru Reed Warbler → IUCN conservation status → Vulnerable → inverse of: IUCN conservation status → Nauru Reed Warbler*; *Reed Warbler → parent taxon → Acrocephalus → inverse of: parent taxon → Nauru Reed Warbler*. While many paths reverse to the seed entity, one of them correctly predicts the first two hops. Since we stop the search at depth two, due to the costs of the implementation (Appendix C.2), this is the best that the model can achieve.

- **SubgraphRAG**: the three ground-truth triples are ranked 5th, 8th, 9th with respect to the relevance scores assigned by the model, hence they are all consistently retrieved.

## E  COMPARISON WITH CONCURRENT WORKS

With the latest advancements in LLM reasoning abilities, strong interest has arisen in using generative AI to create synthetic datasets for a variety of applications (Long et al., 2024). Two concurrent works (Dammu et al., 2025; Zhang et al., 2025) have proposed similar pipelines to the one outlined in Section 3 to create synthetic datasets for KGQA. Here, we discuss differences and reciprocal advantages of these methodologies in detail.

- Dynamic-KGQA (Dammu et al., 2025) is designed to build (question, answer subgraph) pairs on the fly using LLMs, starting from compact seed subgraphs in YAGO 4.5 (Suchanek et al., 2024) that group triples around a common theme. However, as observed in Zhang et al. (2025), the generation pipeline is still prone to hallucinations, with factual correctness of questions estimated to be not better than previous benchmarks. Moreover, no logical query is executed on the KG: the LLM is tasked to judge whether the proposed answer subgraph captures the correct reasoning paths from seed to answer nodes, resulting in higher likelihood of incorrect data, compared to our validation approach based on SPARQL queries.

- KGQAGen (Zhang et al., 2025) proposes a more cost-efficient validation pipeline, based on SPARQL query execution and iterative revisions of incorrect queries (while we directly discard datapoints where the SPARQL query does not execute, or returns incompatible results with the LLM-generated ones). However, it does not keep track of all the seed entities mentioned in the natural-language question, only providing the node from which the seed graph is constructed (equivalent to our entity $s$ in Algorithm 1), which limits the usefulness for benchmarking KG-RAG models that use multiple seed entities at once. The datasets presented in the paper, KGQAGen-10k, contains 10,787 questions with a random 80/10/10 train/validation/test split (while we carefully curate the split of GTSQA to test zero-shot generalization abilities of models), constructed from Wikidata, with seed entities from a set of 16,000 Wikipedia's Vital Articles[4].

While both Dynamic-KGQA and KGQAGen are presented as multi-hop datasets, neither paper includes statistics on the distribution of structures for their respective ground-truth answer subgraphs, making hard to actually evaluate the degree of question complexity in these datasets. On the other hand, our framework makes this very simple, using the metrics provided by the classification of graph isomorphism (Appendix A.2). Similarly, while Dynamic-KGQA uses an LLM as-a-judge to assess the presence of redundant information in the question, our approach based on decomposition of SPARQL queries and identification of minimal subsets of seed entities sufficient to answer (Appendix A.3) uniquely implements an exact measure of redundancy. Finally, GTSQA ensures better reliability as a benchmark for KG-augmented LLMs by checking that the ground-truth subgraphs provided can indeed be utilized by an LLM to answer test questions correctly.

---

[4]https://en.wikipedia.org/wiki/Wikipedia:Vital_articles

