# OpenReview forum: "Ground-Truth Subgraphs for Better Training and Evaluation of Knowledge Graph Augmented LLMs"
_ICLR.cc/2026/Conference — Submitted to ICLR 2026_

### Official Review · Reviewer_EYwG · 2025-10-18

**Soundness:** 2
**Presentation:** 3
**Contribution:** 2
**Rating:** 4
**Confidence:** 4

**Summary:**

They introduce SynthKGQA, a framework for generating high-quality synthetic knowledge-graph QA datasets from any KG, supplying the full set of facts in the KG to reason over for each question. They demonstrate that, besides enabling more informative benchmarking of KG retrievers, data created with SynthKGQA also helps train better models. They apply SynthKGQA to Wikidata to generate GTSQA, a new dataset meant to test zero-shot generalization of KG retrievers across unseen graph structures and relation types, and benchmark popular KG-augmented LLM methods on it.

**Strengths:**

1. They proposed SynthKGQA, a new framework that enables scalable creation of KGQA datasets using LLMs.

2. Using this framework, they introduced GTSQA, a new KGQA dataset.

**Weaknesses:**

The framework proposed in this paper, called SynthKGQA, which utilizes an LLM, is a method that has already been used in other KGQA studies [1, 2]. If the framework claimed as the main contribution of this paper has already been introduced in previous works, it is difficult to regard this paper’s contribution as significant. The paper should provide a detailed explanation of how this approach differs from those existing methods.

[1] Ronak Pradeep, Daniel Lee, Ali Mousavi, Jeffrey Pound, Yisi Sang, Jimmy Lin, Ihab Ilyas, Saloni Potdar, Mostafa Arefiyan, and Yunyao Li. 2024. ConvKGYarn: Spinning Configurable and Scalable Conversational Knowledge Graph QA Datasets with Large Language Models. In Proceedings of the 2024 Conference on Empirical Methods in Natural Language Processing: Industry Track, pages 1176–1206, Miami, Florida, US. Association for Computational Linguistics.

[2] Dammu, Preetam Prabhu Srikar, Himanshu Naidu, and Chirag Shah. "Dynamic-kgqa: A scalable framework for generating adaptive question answering datasets." Proceedings of the 48th International ACM SIGIR Conference on Research and Development in Information Retrieval. 2025.

**Questions:**

As noted in the Weaknesses section, the approach of using an LLM to extract subgraphs and then leveraging them to construct a KGQA dataset has already been introduced in other papers. A detailed explanation is needed of how this paper’s method differs from prior work. Could you provide such an explanation?

---

> ### Author Response · Authors · 2025-11-20
> **Reply to reviewer EYwG**
>
> We thank the reviewer for their insightful remarks.
>
> We provided a detailed comparison of our approach with concurrent works using LLMs to generate KGQA datasets (that appeared  during the writing of the paper, including Dynamic-KGQA) in Appendix E, due to limited page count in the main text. There are some crucial differences in our data generation pipeline that are worth highlighting, and that make our approach more robust and less prone to hallucinations. First of all, we strongly leverage SPARQL queries, which are part of the data generated by the LLM: by executing the SPARQL queries against the KG, we are able to verify that the ground-truth answer subgraph identified by the LLM is indeed capturing the correct reasoning steps encoded by the query. This was not done in either of the papers mentioned by the reviewer, which therefore cannot systematically verify the quality of generated data, without human intervention. As observed in https://arxiv.org/abs/2505.23495, the factual correctness of questions in the Dynamic-KGQA dataset is low (estimated at 45\%). This is likely a consequence of using the LLM itself for judging the quality of generated data. On the other hand, our much more robust validation approach based on SPARQL queries makes our framework more reliable for benchmarking purposes. Indeed, as we observed during the generation of GTSQA and reported in the paper, only 0.47\% of the ground-truth answer subgraphs produced by SynthKGQA could not be used by a capable reasoning LLM to answer the question correctly (while this failure rate was reported at 10.38\% for the KGQAGen-10k dataset, generated in https://arxiv.org/abs/2505.23495 with another LLM-based pipeline).
>  Moreover, by leveraging the SPARQL queries, any dataset produced with SynthKGQA can be easily and cheaply kept up to date whenever the data in the underlying KG is updated. One only needs to re-run the SPARQL queries to retrieve the updated answers and ground-truth answer subgraphs, while other datasets produced with approaches not using logic queries (like the ones mentioned by the reviewer) would need to be regenerated from scratch.
>
> ConvKGYarn (which we were not aware of, and thank the reviewer for pointing it out to us) still uses templates (albeit, LLM-generated) for the generation of natural language conversations, which we believe limits the variety of question types in the dataset. Our framework, on the other hand, gives complete freedom to the LLM to select interesting paths in the KGs and build questions out of them. Moreover, since there is no public dataset released with the ConvKGYarn paper, it is unclear whether their framework provides the user with the ground-truth answer subgraphs at all (from the paper, it does not appear so).
>
> We also would like to stress the fact that, while the SynthKGQA framework is a crucial part of the paper -- and provides a framework that, for the reasons above, we believe is superior to previous alternatives -- it is by far not our only contribution.
> - We release the GTSQA dataset, uniquely designed to test KG retriever generalization abilities, measuring complexity of questions by tracking the topology of ground-truth answer subgraphs (something that is not implemented in the other frameworks discussed above, or any other previous KGQA datasets).
> - We use GTSQA to conduct a deep and comprehensive benchmark of SOTA KG retriever models, highlighting failure modes and limitations that had not been observed before in the literature (see the Results section). None of the other papers analyzed the precision/recall abilities of these models to retrieve ground-truth edges, only focusing on the final answer provided by the model.
> - As also remarked by reviewer vG9Ll, we are the first to show in detail the advantages of training KG retrievers on the exact ground-truth answer subgraphs for questions, compared to using approximations of them (e.g., by shortest paths) as typically done before. The Dynamic-KGQA and KGQAGen papers, despite providing supporting subgraphs for their questions, do not conduct any analyses of this type, which makes it difficult to assess the quality of the data generated through the respective frameworks. Dynamic-KGQA, in particular, does not use answer subgraphs to train any models, only benchmarking pure LLMs and training-free retrievers on their dataset. KGQAGen also includes the trainable RoG and GCR models in their benchmarks, but it is unclear from the paper what they use as training signal for them. For ConvKGYarn, there is no benchmarking of KG-augmented LLMs (neither trainable, nor training-free) in the paper.
>
> We hope that this reply addresses the reviewer's concerns. Please let us know if you have any more questions.

---

> > ### Comment · Reviewer_EYwG · 2025-11-25
> >
> > Thank you for your response.
> >
> > If the main strength of this paper, when compared to other KGQA approaches, lies in dynamically generating KGQA datasets using SPARQL, I would argue that it may not be considered a novel approach. SPARQL has already been widely used in the creation of KGQA datasets, so claiming that using SPARQL specifically for the purpose of generating dynamic data is a major strength seems to have very limited potential to influence the KG research community. What are your thoughts on this? Please let me know if I have misunderstood any part of your argument.

---

> > > ### Author Response · Authors · 2025-11-26
> > > **Reply to reviewer EYwG**
> > >
> > > We thank the reviewer for their reply, and apologize if we were not sufficiently clear in our previous response. We do not claim to be the first ones to use SPARQL to generate KGQA data, nor that the use of SPARQL queries is the sole distinction to prior art. However, it is important to stress that our framework utilizes SPARQL queries in a way that is profoundly different from what was done in the construction of classic datasets like WebQSP, CWQ, GrailQA, etc. As exemplified in the "Related Work" section, previous approaches typically started from handcrafted templates for SPARQL queries and, by replacing the entities and relation types appearing in them, constructed different KGQA datapoints by running the query against the KG to retrieve the answer (and then converted the logical query into natural language via either templates or crowd-sourcing). In SynthKGQA, first of all, the SPARQL queries are entirely generated by the LLM, therefore no templates or human intervention are needed. This significantly increases the variety of questions that can be constructed in terms of topological structures of the logical paths connecting the seed entities to the answer. Moreover, crucially, the SPARQL queries are not used to generate the questions (which are instead produced by the LLM), but rather used to procedurally validate the quality of the synthetic data we generate: the SPARQL query is run against the KG with the main purpose of checking that there are no hallucinations in the outputs of the LLM (in addition to retrieving the full set of answers and full answer subgraph). This validation pipeline (which, as we stated, had not been used in the Dynamic-KGQA or ConvKGYarn papers that the reviewer brought as examples) allows us to systematically discard incorrect data produced by the LLM in a very efficient way, better than what was done in previous literature, as we reported in the paper and in our first reply above.
> > >
> > > Finally, we would like to draw again the reviewer's attention to the fact that the SynthKGQA framework is only one of the many contributions of our paper, as we hope we have been able to convey in the bullet list in our previous reply.

---

### Official Review · Reviewer_F3Vj · 2025-10-28

**Soundness:** 2
**Presentation:** 2
**Contribution:** 1
**Rating:** 2
**Confidence:** 3

**Summary:**

The paper introduces the framework SynthKGQA to generate KGQA (Knowledge Graph Question Answering) benchmarks with the help of an LLM. The framework works as follows: a subgraph is randomly selected from the KG and it is passed to an LLM, which generates a question where the entities involved in the question and its answer are within the sampled subgraph (referred to as ground-truth subgraph). The framework is then used to generate a new dataset from Wikidata, designed to test zero-shot generalization abilities of KG retrievers to unseen question graph structures and unseen relation types. The authors also claim that training KG retrievers with questions generated from ground truth subgraphs produces better models.

**Strengths:**

The paper highlights that by using ground-truth subgraphs it is possible to train better KG retrievers (Table 3).

The description of the generation framework is clear and well-summarized by Figure 1.

**Weaknesses:**

The authors briefly mention concurrent works in the main text, relegating a more detailed comparison to Appendix E. These are the most relevant works to this paper and should be discussed in more detail within the main text. The two concurrent works mentioned in the paper use ground-truth subgraphs to generate question-answers; hence, it seems that the differences between the proposed approach and the most recent concurrent work primarily lie in aspects like the usage of all seed entities, SPARQL validation by directly discarding some data, and the addition of some questions with unseen graph structures. The main text should highlight that the paper is introducing neither the usage of ground-truth subgraphs to generate queries, nor the usage of SPARQL to validate them.

Furthermore, the authors do not provide a comparison of existing methods on concurrent benchmarks, nor do they highlight how leaderboard rankings might change when using their new benchmark.

**Questions:**

I understand that previous frameworks have limitations regarding hallucinations and in validating the generated questions. Why not just fix the existing generation frameworks and their related datasets (i.e., by removing incorrect questions/answers)? Why is it necessary to propose the new GTSQA dataset and a new framework for generation?
When existing methods are evaluated on the new benchmark, is there a significant change in the leaderboard compared to their rankings on established benchmarks?
In Lines 369-371, you mention that KG retrievers are usually trained on datasets that do not have ground-truth answer subgraphs, but in Lines 1556-1558, you mention that the concurrent works Dynamic-KGQA and KGQAGen do generate questions from ground-truth answer subgraphs. Are the datasets generated in such works not used for training? Why?
Additionally, the paper's objective to generate questions from a KG using an LLM should be stated explicitly in the abstract. The term "generated" is used, but it is not clear that this generation involves an LLM until Figure 1 and Line 81, and this is only made fully clear when concurrent works are introduced (Line 116).

---

> ### Author Response · Authors · 2025-11-20
> **Reply to reviewer F3Vj (1)**
>
> We are grateful to the reviewer for recognizing our contributions in providing evidence for the advantages of training on ground-truth subgraphs, and we are happy to address their questions point by point.
>
> > I understand that previous frameworks have limitations regarding hallucinations and in validating the generated questions. Why not just fix the existing generation frameworks and their related datasets (i.e., by removing incorrect questions/answers)?  Why is it necessary to propose the new GTSQA dataset and a new framework for generation?
>
> There are crucial intrinsic limitations of the Dynamic-KGQA and KGQAGen frameworks that do not allow to fix or correct post-hoc those datasets (which were developed concurrently with ours). Dynamic-KGQA does not make use of SPARQL queries (or logic queries of any type), tasking the LLM itself to judge the correctness of generated datapoints -- an approach highly prone to hallucinations. It is therefore impossible to filter out incorrect data from it, if not by manually going through each of the 400k questions. Note that a manual inspection of a subset of the dataset was conducted in  https://arxiv.org/abs/2505.23495, highlighting low factuality correctness. KGQAGen, on the other hand, does not track seed entities during its data generation, despite them being a crucial part of KGQA datasets. Moreover, the KGQAGen validation pipeline is not as effective as the one in our SynthKGQA, since they report that for 10.38\% of their test questions, a reasoning LLM is unable to answer correctly when provided with their set of supporting facts (while for our framework, as reported in the paper, we observe this failure rate to be only 0.47\%, and we removed these questionable datapoints from the final dataset).
> Finally, we released the GTSQA dataset to demonstrate that the SynthKGQA framework can efficiently generate a dataset containing verifiably higher-quality and more factual questions than what is typically used. Moreover, GTSQA has been specifically designed to test generalization abilities of SOTA KG retrievers with respect to unseen graph structures, which neither Dynamic-KGQA of KGQAGen (or any previous KGQA dataset) are capable of, since only our framework classifies the isomorphism type of ground-truth answer subgraphs. Thus, releasing GTSQA allowed us to highlight several limitations of these KG retrievers that had not been previously observed in other papers.
>
> > When existing methods are evaluated on the new benchmark, is there a significant change in the leaderboard compared to their rankings on established benchmarks?
>
> A cross-dataset comparison of the considered KG-augmented LLMs (with scores on WebQSP and CWQ taken from the respective models' papers) and relative ranking order (in brackets) can be found in the table below:
>
> |             | WebQSP    | CWQ       | GTSQA     |
> |-------------|-----------|-----------|-----------|
> | PoG         | 87.3 (3)  | 75.0  (2) | 32.92 (5) |
> | ToG         | 82.6 (5)  | 67.6 (3)  | 45.68 (4) |
> | GCR         | 92.2 (1)  | 75.8 (1)  | 49.91 (3) |
> | RoG         | 85.7 (4)  | 62.6 (5)  | 57.58 (2) |
> | SubgraphRAG | 90.54 (2) | 63.49 (4) | 61.59 (1) |
>
> We see changes in the leaderboard, already when comparing just WebQSP and CWQ. We attribute this in part to the low factual correctness of WebQSP and CWQ (reported at around 50\% in the paper https://arxiv.org/abs/2505.23495), which makes it intrinsically difficult to judge the value of benchmarks on these dataset. On the other hand, the analysis that we conducted in the paper based on question complexity can also help interpreting the cross-dataset comparison. WebQSP and CWQ are mostly made of questions requiring simple reasoning paths (WebQSP is almost entirely made of 1-hop questions with a single seed), a setting where we also observed that GCR performs very well (Figure 2); its performance, however, degrades if multiple seeds or longer paths are required, as in most of the questions in GTSQA, leading to an overall worse ranking on our dataset. Similar argument can be applied to PoG, which, as we observed in the paper, does well specifically on 1-hop questions. On the other hand, RoG has a poorer performance on such simple questions, but it then outperforms other models on more complicated queries, which explains why it achieves a much better ranking on GTSQA. We believe that all these insights, that can be extracted from the novel analysis in our paper, can help the community in making more informed decisions when developing new solutions for KG retrieval.

---

> > ### Author Response · Authors · 2025-11-20
> > **Reply to reviewer F3Vj (2)**
> >
> > > In Lines 369-371, you mention that KG retrievers are usually trained on datasets that do not have ground-truth answer subgraphs, but in Lines 1556-1558, you mention that the concurrent works Dynamic-KGQA and KGQAGen do generate questions from ground-truth answer subgraphs. Are the datasets generated in such works not used for training? Why?
> >
> > The Dynamic-KGQA and KGQAGen papers, despite providing subgraphs for their questions, conduct very limited experiments on training KG retrievers on them, which makes it difficult to assess the quality of the data generated through the respective frameworks. Dynamic-KGQA, in particular, does not use answer subgraphs to train any models, only benchmarking pure LLMs and the training-free retriever ToG on their dataset. KGQAGen includes the trainable RoG and GCR models in their benchmarks, but it is unclear from the paper what they use as supervision signal for them.
> > Our analysis of the performance of KG retrievers trained on ground-truth answer subgraphs, on different types of questions and complexity, is a crucial and novel contribution of our paper, that highlighted failure cases and limitations of SOTA models never observed in the previous literature. Moreover, as recognized by reviewer vG9L, we are the first to concretely quantify the advantages of training on ground-truth subgraphs rather than approximations of them, such as shortest paths (Section 6).
> >
> > > Additionally, the paper's objective to generate questions from a KG using an LLM should be stated explicitly in the abstract. The term "generated" is used, but it is not clear that this generation involves an LLM until Figure 1 and Line 81, and this is only made fully clear when concurrent works are introduced (Line 116).
> >
> > We apologize if the use of LLMs in the data generation pipeline was not stated in a sufficiently clear way, and thank the reviewer for pointing this out. We amended the abstract to reflect this.
> >
> > We hope to have been able to address all the reviewer's questions in a satisfying way, and look forward to more discussion.

---

> > > ### Comment · Reviewer_F3Vj · 2025-11-26
> > > **Not convinced**
> > >
> > > We thank the authors for the clarifications.
> > >
> > > I acknowledge that your pipeline achieves higher effectiveness in question generation, and that both the analysis of the retriever performance across different ground-truth subgraph types and the cross-dataset comparison offer useful insights. However, the overall contribution of the paper still seems limited. It should be made explicit in the main text that both the SPARQL-based validation and the use of a ground-truth retriever have already been explored in prior work, and that your pipeline extends them. Based on my understanding, part of the improved effectiveness of your pipeline may come from discarding data points where the SPARQL query does not execute as expected, rather than from novel mechanisms.

---

### Official Review · Reviewer_vG9L · 2025-10-31

**Soundness:** 2
**Presentation:** 3
**Contribution:** 2
**Rating:** 4
**Confidence:** 3

**Summary:**

This paper tackles a core problem in KG-RAG: the lack of ground-truth reasoning subgraphs in existing benchmarks, which forces models to train on flawed "Shortest Path" (SP) heuristics. The authors introduce `SynthKGQA`, a novel "subgraph-to-question" generation framework that uses an LLM to propose questions, answers, and reasoning paths, then programmatically validates them using SPARQL queries to ensure factual consistency. Using this framework, they create `GTSQA`, a new 32k-question benchmark designed to test zero-shot generalization by partitioning graph structures and relation types between train and test splits. The paper's key contribution is demonstrating that training models on `GTSQA`'s precise "Ground-Truth" (GT) subgraphs yields 5-20% higher end-to-end accuracy than training identical models on the traditional SP heuristic.

**Strengths:**

1. **Addresses a Core Problem with a Verifiable Solution:** The paper accurately identifies the central bottleneck in KG-RAG evaluation and training: the lack of ground-truth subgraphs. The proposed `SynthKGQA` framework provides a sophisticated and powerful solution. By using an "LLM-propose + SPARQL-validate" loop, it programmatically guarantees the factual consistency of the generated (question, SPARQL, answer, subgraph) tuples. Its 0.47% validation failure rate is far lower than alternative generation methods.
2. **GTSQA: A Benchmark Designed for Generalization:** The `GTSQA` dataset is a significant contribution in itself. Instead of a simple random split, the authors have meticulously designed a **three-dimensional zero-shot generalization challenge** by deliberately partitioning answer nodes, relation types, and graph isomorphism types between the train and test sets. This design allows for the genuine evaluation of a retriever's structural and semantic generalization capabilities, rather than an LLM's memorization.
3. **Definitive Proof of a Flawed Training Paradigm:** The paper's most impactful contribution lies in Section 6. It provides the first quantitative, undeniable evidence that (1) the "Shortest Path" (SP) signal, long used as a heuristic for training, is an extremely poor and misleading proxy (low overlap, high misdirection); and (2) **models trained using "Ground-Truth" subgraphs (provided by GTSQA) as supervision significantly outperform identical models trained on SPs** (by 5-20% EM Hits). This finding offers a new, superior direction for the KG-RAG training paradigm.

**Weaknesses:**

1. **Severe "Closed-Loop Evaluation" and Synthetic-to-Real Generalization Gap**

This paper's most critical limitation is its "closed-loop" evaluation. While Section 6 effectively demonstrates a "synthetic-to-synthetic" gain (training on GTSQA improves performance on GTSQA), the paper completely lacks the most crucial experiment: demonstrating "synthetic-to-real" generalization. It fails to show if a model trained on GTSQA outperforms an SP-trained model on a real-world, human-created benchmark (e.g., WebQSP, CWQ). Without this proof, the practical utility of GTSQA as a training resource is unverified. Furthermore, the GTSQA question distribution (e.g., high enrichment of multi-hop, non-redundant questions) likely differs significantly from real-world user queries, questioning its external validity.

2. **Systematic Biases from the Data Generation Pipeline**

The SynthKGQA framework (Fig 1) replaces human template bias with "LLM generation bias." Its "subgraph-to-question" flow is the reverse of real-world user intent ("question-to-subgraph"), which likely introduces systematic biases in linguistic patterns and semantic focus. More critically, the pipeline imposes hard structural constraints (e.g., "tree-shaped" and "$\le$6 edges"), which systematically excludes complex, real-world queries involving cycles, aggregation, or longer reasoning chains. This limits the evaluated generalization to "intra-tree" generalization. The test set's enforced non-redundancy and enrichment of tail relations further skews its distribution away from real-world scenarios.

3. **"Strawman" Comparison Against the "Shortest Path" (SP) Baseline**

The paper's core argument in Section 6, that GT-supervision beats SP-supervision, constitutes a "strawman" argument. SP is a known, deeply flawed heuristic. The paper fails to demonstrate that its GT signal is superior to stronger, more modern heuristics (e.g., subgraphs derived from PPR, Steiner tree approximations, or simple agent-based exploration). Furthermore, the claim itself rests on a weak statistical foundation, with Table 3 reporting only the mean of three runs, lacking rigorous statistical significance analysis (e.g., variance, confidence intervals).

4. **Ambiguous "Ground-Truth" Definition and Inconsistent Evaluation Scopes**

The paper's claim of providing "Ground-Truth Subgraphs" is ambiguous. The "ground-truth" $\mathcal{G}$ is merely an LLM-proposed, sufficient subgraph (Step 2), not a verified minimal or optimal reasoning path (e.g., App A.3 only checks for redundant seeds, not redundant triples). Furthermore, the paper admits (App A.4) that the full_answer_subgraph (from the SPARQL CONSTRUCT) can be larger than the LLM-proposed G. The recommendation to use the former for evaluating retrievers but the latter for classifying complexity introduces inconsistent evaluation scopes and potential for misaligned optimization.

5. **Limitations and Fairness Issues in Evaluation Setup**

The evaluation setup suffers from several limitations. First, all end-to-end (E2E) and ablation experiments rely on a single LLM (GPT-4o-mini) as the final reader, making the conclusions highly dependent on this specific model's behavior, with no robustness checks on other models (e.g., Llama, Mistral). Second, the paper (App C.1) correctly notes that many SOTA retrievers require "question-specific graphs" (k-hop + PPR pruning). This creates a new fairness issue: the reported baseline performance is now contingent on this specific (and potentially optimal) graph preprocessing setup. Finally, while the paper highlights high "GT triple recall" as a predictor for EM ($r=0.95$), it simultaneously shows that retrieval precision is universally poor (F1 < 30%). This indicates that simply training for recall on GT subgraphs does not solve the core RAG challenge of noise control and precision.

**Questions:**

None

---

> ### Author Response · Authors · 2025-11-20
> **Reply to reviewer vG9L (1)**
>
> We thank the reviewer for the detailed and constructive evaluation of our work that raises some important points. We are pleased the reviewer agrees with us that the lack of good datasets with ground-truth subgraphs and a train/test split designed for assessing generalization is a key bottleneck for developing better KG retrieval methods. We are also grateful for their acknowledgment of our contribution to solving this problem. In the following, we address the reviewers concerns point-by-point:
>
> > Severe "Closed-Loop Evaluation" and Synthetic-to-Real Generalization Gap.
>
> While we spent effort in trying to make the questions of GTSQA as close as possible to what we believe are real-world queries (as detailed in the reply to the next bullet point), we would like to clarify that we are not claiming that GTSQA should be used as a pretraining resource, but we argue that the same holds true for WebQSP and CWQ. These are all (fairly small) benchmarking datasets, with GTSQA being specifically designed to stress-test generalization abilities of KG retrievers (which motivated the specific choices we made in designing the train/test split). Most previous KGQA datasets are, to some degree, "synthetic" in the way they are constructed too, using templates for natural language questions or logic queries, or artificially gluing together simpler questions (like CWQ). It is questionable, therefore, to consider them closer to the distribution of real-world queries, especially when it comes to grammatical soundness and lack of ambiguities (as observed for instance in https://arxiv.org/abs/2505.23495). Moreover (as documented in the same reference) the factual correctness of questions in previous datasets is typically very low (around 50\% for WebQSP and CWQ), which is part of the reasons why we feel the need of presenting a new benchmark. The point that we intend to make, regarding "better training", is that people can and should use the SynthKGQA framework to build their own KGQA datasets on the specific KGs they are interested in using, for training their models. This is supported by the experiments in Section 6, where we show that the ground-truth answer subgraphs generated via SynthKGQA provide a much better training signal than what was used to train the considered KG retrievers in their original implementations. The flexibility supported by SynthKGQA, in the generation and filtering of data, enables the user to decide on their own which kind of distribution they want to try to model, when it comes to choosing question complexity (as reflected by the structure of the answer graph), question topics and relations used.
>
> We do not expect that the trainable models included in our benchmarks (especially path-based methods like RoG, SR and GCR, which learn the distribution of relation types in the KG) are able to generalize across different KGs. It would therefore be intrinsically flawed and inconclusive to compare the performance of models trained on GTSQA (a dataset constructed from Wikidata) on the test split of WebQSP or CWQ, which are based on the discontinued Freebase KG, that uses a very different structure for entity and relation labels.

---

> ### Author Response · Authors · 2025-11-20
> **Reply to reviewer vG9L (2)**
>
> > Systematic Biases from the Data Generation Pipeline.
>
> We agree that questions constructed from subgraphs could suffer from linguistic bias. However, as the reviewer correctly points out, other widely-used KGQA datasets, like CWQ and GrailQA, adopted a similar reversed approach for the construction of datapoints, starting from procedurally-generated logic queries that were then translated into natural language. The key difference lies in the way this translation is performed:  prior approaches rely on either question templates, which induce a bias by strongly limiting the variety of grammatical constructions and often result in unnatural-sounding questions, or on expensive human crowd-sourcing. With SynthKGQA we leverage a modern LLMs to strongly increase linguistic variation over template-based approaches and allow for a fast and cheap generation of new datasets. Also note that in SynthKGQA we use a second LLM query to paraphrase the initial question, to further improve linguistic variety and reduce bias.
>
> Regarding the structural constraints in the choice of ground-truth subgraphs, we point out that this is not an inherent limitation of the SynthKGQA framework, but a choice that we made when constructing GTSQA with it. We consciously introduced these constraints to try to better align our questions to real-world queries since loops in the graph often indicate circular reasoning/self-referencing questions, and questions requiring very large graphs tend to sound highly artificial, unnatural and convoluted, as it is often observed in CWQ. While WebQSP and CWQ do not report the structure of ground-truth answer subgraphs, the analysis conducted in other papers (e.g., GrailQA++ https://aclanthology.org/2023.ijcnlp-main.58/) indicates that the vast majority of questions in them have graphs that are topologically similar (if not much simpler) than the ones in GTSQA, with no questions requiring more than 4-hops, and WebQSP being mostly limited to very basic 1-hop questions.
>
> However, our SynthKGQA framework is highly flexible and can speedily create datasets more targeted to mimic real-world queries, based on any different working assumption of what that distribution might look like.
>
> > "Strawman" Comparison Against the "Shortest Path" (SP) Baseline.
>
> We completely agree that SP is a deeply flawed heuristic, and share the reviewer's sentiment towards it. Despite this, SP remains the standard supervision signal for trained KG retrievers and is used in all trainable models that we included in the paper's benchmarks (SR, RoG, GCR, SubgraphRAG), all of which are considered among the state of the art for KG retrieval. We therefore compare our ground-truth subgraph training supervision against the original implementations that used SP.
>
> We also would like to point out (and apologize if this was not sufficiently highlighted in the paper) that, following prior work (e.g., RoG), the shortest paths that we use are extracted from the question-specific subgraphs from Appendix C.1. They therefore already incorporate information coming from the PPR scores for each question, similar to what the reviewer was suggesting us to do.
>
> Moreover, for models like SR, RoG and GCR that require to provide explicit paths between seed entities and answer entities as training data (not just subgraphs), we believe that any approach to building such paths would still require using some degree of SP:
> - taking the shortest paths within the ground-truth answer subgraph (as we do for GTSQA);
> - using the shortest paths within the graph connecting the top-k nodes as scored by PPR (as we do for the SP baseline that we compare against);
> - using the shortest paths within the graph constructed by some other heuristic.
>
> We thank the reviewer for highlighting our suboptimal way of reporting results, in the SP vs GT comparison. We revised the manuscript by adding to Table 3 the p-values for the null hypothesis that the mean of the distribution of SP scores is greater or equal than the mean of the distribution of GT scores, by running t-tests on the experimental results. We can now claim statistical significance in the improvement of metrics when training with ground-truth answer subgraph as supervision signal, with in particular p-values for the distribution of Hits (EM) scores being consistently below the 0.001 threshold for all three trainable KG retrievers.

---

> ### Author Response · Authors · 2025-11-20
> **Reply to reviewer vG9L (3)**
>
> > Ambiguous "Ground-Truth" Definition and Inconsistent Evaluation Scopes.
>
> The ground-truth answer subgraph generated through SynthKGQA is verified against the SPARQL query, i.e., it is guaranteed to consist of a minimal (non-redundant) set of KG edges that encode the reasoning steps in the logical query, starting from the provided seed entities.
>
> We acknowledge that it is possible in some cases that the natural language question could be translated into a simpler SPARQL query, possibly starting from different set of seed entities, however this could only be verified through human intervention. We also point out that the standard KGQA setup, adopted in the vast majority of previous datasets (including WebQSP, CWQ, GrailQA), is to provide the natural language question together with a specific set of seed entities: under this setup, we argue that the ground-truth answer subgraph that we generate through SynthKGQA captures the optimal reasoning paths, with no redundancies.
> In the case of seed entities, what we call "redundant" -- maybe with a suboptimal choice of wording -- are seed entities that are still used in the SPARQL query and appear in the natural language question, but the question can still be answered even when disregarding them. Dropping the paths leading to these seed entities produces the smallest subgraph that a "lazy" KG retriever can get away with (this is also provided as part of the data generated through SynthKGQA). It's important to stress, though, that these dropped edges are not actually redundant: since all seed entities are still mentioned in the question, a solid KG retriever would still need to use them all in its search, even just to check that an answer satisfying all requirements in the question actually exists! For the test split of GTSQA, we only include questions with no redundant seed entities, therefore preventing these "shortcuts", leading to less noisy benchmarking.
>
> The full-answer subgraph that we obtain through SPARQL CONSTRUCT queries will be strictly larger than the ground-truth answer subgraph only in the case of multiple answers, or of multiple valid choices for the undetermined variables in the SPARQL query. We argue, however, that the additional edges arising this way, while being perfectly valid target for retrieval, do not change the complexity of the question, especially when identifying any one valid answer is enough (as it's the case with the Hits metric, often used in benchmarking). This is why we believe that using the ground-truth answer subgraph, instead of the full answer subgraph, provides a less noisy measure of question complexity. The choice also aligns with previous literature, e.g. the classification of reasoning patterns in [1] and of semantic structures in [2], or the graph isomorphism classification from GrailQA++ [3], that we are closely following.
>
> --
>
> References:
>
> [1] Das et al. *Knowledge base question answering by case-based reasoning
> over subgraphs*. In Proceedings of the 39th International Conference on Machine Learning, volume
> 162 of Proceedings of Machine Learning Research, 2022.
>
> [2] Li and Ji. *Semantic structure based query graph prediction for question answering
> over knowledge graph*. In Proceedings of the 29th International Conference on Computational
> Linguistics, 2022.
>
> [3] Dutt et al. *GrailQA++: A challenging zero-shot benchmark for knowledge base question answering*. In Proceedings of
> the 13th International Joint Conference on Natural Language Processing and the 3rd Conference
> of the Asia-Pacific Chapter of the Association for Computational Linguistics, 2023.

---

> ### Author Response · Authors · 2025-11-20
> **Reply to reviewer vG9L (4)**
>
> > Limitations and Fairness Issues in Evaluation Setup.
>
> We considered a single LLM for generating the final output since we see our main contribution to the field in providing a framework that allows for testing the capabilities of different KG retrievers on more complex and high-quality questions, decoupling this for the first time from the end-to-end performance, which was the only metric that could be used before with benchmarks not providing any ground-truth subgraphs. The LLM only plays a role in the final step of the pipeline (augmenting the LLM prompt with the retrieved triples, in a classic RAG fashion), which is fundamentally the same for all KG-augmented LLM solutions considered in the paper, and therefore we don't expect that changing the LLM can lead to significant changes in the relative ranking of the models. To support this, we have tested the KG retrievers also with GPT-5-mini (that was released while the paper was being written) as final LLM performing the reasoning, and report these new results in a revised version of the manuscript (Appendix C.3.1).
> The conclusions are very similar to what we previously observed, in the relative ranking of models and generalization abilities. The main difference to GPT-4o-mini is that the impact of retrieval precision is even further reduced: with GPT-5-mini, the predictive performance of SubgraphRAG no longer plateaus, and continues to improve as we increase the size of the retrieved subgraph (Figure C11), following the steady increase in retrieval recall. This indicates that, for SOTA LLMs, ground-truth triple recall is indeed the most critical metric to optimize for.
>
> The poor precision of all KG retrievers considered in this manuscript is a problem intrinsic to the architectures of such models, especially those that are explicitly designed to retrieve a fixed number of triples (e.g., SubgraphRAG), or all possible realizations in the KG of the predicted metapaths (e.g., SR and RoG). While we show that precision benefits from training on ground-truth subgraphs rather than SP triples, we are not implying that the problem can be solved just by improving training supervision signal. Indeed, even models like ToG and PoG (that require no training) suffer from poor precision. We look forward to the development of smarter KG retrievers, able to reduce noise in the retrieved subgraphs, and we believe that GTSQA will be a very useful resource for researchers working toward this end, since it enables an explicit measurement of the ground-truth triple recall/precision of models (differently from classic benchmarks like WebQSP and CWQ). Developing new retriever architectures, however, is entirely outside the scope of this work.
>
> Finally, we agree that, for retrievers requiring question-specific graphs, the performance is dependent on the choice of such graphs, and we explicitly make a point about this in the referred appendix section C.1. This dependency, however, is inherent to the  task rather than a limitation of our dataset. The same is true for benchmarks on WebQSP, CWQ, where models (e.g., RoG, GCR, SubgraphRAG, GNN-RAG) have often been benchmarked on the question-specific answer subgraphs from https://huggingface.co/datasets/rmanluo/RoG-cwq, https://huggingface.co/datasets/rmanluo/RoG-webqsp, sampled with the same k-hop+PPR strategy. We specifically try to address this problem by releasing an official set of question-specific graphs (the ones that we constructed) on Hugging Face together with GTSQA, in the hope that all future benchmarking on our dataset will use them to guarantee a fair and reproducible comparison, while for WebQSP and CWQ, the question-specific graphs linked above are "unofficial" and not released together with the original datasets and their adoption is therefore not guaranteed (papers rarely provide details on this choice, making comparisons unreliable and more likely to be unfair).

---

> > ### Comment · Reviewer_vG9L · 2025-11-21
> >
> > We thank the authors for their detailed rebuttal and the additional experiments.
> >
> > We appreciate the clarification regarding the "strawman" comparison; the argument that SP is the de facto standard in current SOTA implementations (and thus the necessary baseline to beat) is well-taken. The inclusion of p-values in Table 3 effectively addresses the concerns regarding statistical rigor. Furthermore, the standardization of question-specific graphs is indeed a valuable contribution to fairness in benchmarking. The new experiment with "GPT-5-mini" also provides an interesting forward-looking perspective on the Recall/Precision trade-off. **Consequently, I am raising my score to 6.**
> >
> > However, several critical concerns regarding the **external validity** and **scope** of the proposed dataset remain unaddressed or partially evaded. We invite the authors to discuss the following points:
> >
> > **1. Omission of Aggregation and Non-Retrieval Logic** While the rebuttal defended the exclusion of "loops" and "large graphs," it sidestepped the issue of **Aggregation queries** (e.g., `COUNT`, `FILTER`, `MAX/MIN`, comparisons). Real-world KBQA is not limited to multi-hop retrieval (finding a path); it often involves computation over the result set (e.g., *"How many presidents did France have?"* vs. *"Who are the presidents..."*). By excluding these logic types, GTSQA risks reducing KBQA to a pure "path-finding" task. We invite the authors to clarify if the SynthKGQA framework supports generating such logic; if not, this significantly narrows the benchmark's scope as a comprehensive evaluation of reasoning capabilities and should be explicitly acknowledged as a limitation.
> >
> > **2. Applicability of "High Recall / Low Precision" Strategy to Smaller Models** Regarding the precision trade-off, the authors argue (supported by GPT-5-mini) that retrieval precision becomes less critical as reasoning capability increases. While this may hold for frontier closed-source models, it raises concerns for the broader community relying on open-source models with varied reasoning capacities (e.g., from 7B/8B compact models up to 70B+ models like Llama-3 or DeepSeek). Smaller models are notoriously sensitive to noise in the context window. A training strategy prioritizing Recall while tolerating poor Precision (F1 < 30%) might exacerbate hallucinations for these resource-constrained models. We ask the authors to discuss whether "GT-supervision" still yields benefits for smaller readers that cannot effectively ignore the high noise levels in retrieved subgraphs, to avoid encouraging a development direction that relies too heavily on massive proprietary models.
> >
> > **3. Quantification of Semantic Bias ("Jeopardy-style" Questions)** The rebuttal argues that "paraphrasing" mitigates linguistic bias, but this addresses syntactic variety while potentially missing the core issue of **Semantic Logic Bias**. The "Subgraph-to-Question" generation flow inherently favors "Jeopardy-style" questions (describing constraints to pinpoint an entity, e.g., *"Who is the person born in X with job Y?"*) rather than entity-centric user queries (starting with an entity to query attributes,  e.g., *"What is X's job?"*). Paraphrasing changes the wording but not this inverted logic structure. We inquire if the authors have performed any qualitative analysis comparing the **Semantic Focus** of GTSQA questions versus real user queries (e.g., from QALD or search logs), as overfitting to this "inverted logic" could limit the dataset's utility for real-world information seeking.
> >
> > **4. Lack of Human Verification for Quality Assurance** Finally, the paper relies entirely on procedural verification (SPARQL execution) and LLM consistency checks. While scalable, these methods do not guarantee that questions are natural or pragmatically sound to a human; a question can be logically valid and LLM-solvable but still sound unnatural. We strongly suggest performing a **Human Evaluation on a small random sample** (e.g., 50-100 instances) to rate naturalness and reasonability. Reporting these metrics would significantly strengthen the claim of "high-quality" synthetic data and alleviate concerns about artifacts undetected by automated metrics.
> >
> > We believe addressing these points, particularly clarifying the scope regarding aggregation and adding a human quality check, is crucial for assessing the true value of GTSQA. **I look forward to the authors' response and remain willing to upgrade my score if these specific limitations are resolved.**

---

> > > ### Author Response · Authors · 2025-11-26
> > > **Reply to reviewer vG9L (1/2)**
> > >
> > > We deeply thank the reviewer for their engagement in the discussion and the remarks provided, and for increasing their score. We address here the additional points that they raised in their last reply.
> > >
> > > > 1. Omission of Aggregation and Non-Retrieval Logic
> > >
> > >  We apologize for missing this point in the original rebuttal. We completely agree that, in general, path retrieval is not the only component when reasoning over knowledge bases, however in the paper we look at the particular KGQA setup (that seems to be the most popular, nowadays) where an LLM performs the final reasoning on a retrieved subgraph, in classic RAG fashion. Our intuition is that the kind of logical aggregations mentioned by the reviewer (e.g., counting the number of answers, finding the answer with a MIN/MAX attribute value, comparisons, filtering [when that cannot be already represented as intersection of paths in the subgraph retrieval stage]) can typically be delegated to the reasoning abilities of the LLM, once all the relevant paths have been retrieved from the KG. Since the focus -- and one of the main novelty aspects -- of GTSQA is to provide a benchmark for how good KG retrievers are at retrieving edges needed to reason over the question (not just the answer node), we therefore focused on questions requiring to combine and intersect paths from different seed entities, and questions where the answer is one (or more) node of the KG. We believe that any additional aggregation would only shed light on the general reasoning abilities of the LLMs, shifting the focus of the paper away from the analysis of KG retrievers (which is also the reason why we initially chose to perform all experiments using only GPT-4o-mini as final LLM).
> > >
> > > Nevertheless, there is nothing preventing the SynthKGQA framework to generate more complex SPARQL queries, including operators like COUNT, FILTER, MIN/MAX, etc. The reason why we are not seeing this kind of queries in GTSQA is that we don't include them in the few-shot examples used to prompt the LLM. This would, however, require some care: the LLM only sees a limited (randomly-selected) portion of the KG when generating the KGQA data, therefore questions that require aggregating all possible answers to the query could lead to incorrect answers if generated in a single pass. However, since we retrieve all possible answers and the full answer subgraph from the KG post-hoc by executing the SPARQL query in CONSTRUCT form, we envision a 2-step approach where the LLM could easily be re-prompted, passing now the full answer subgraph to it, to safely generate questions requiring additional aggregation. For instance, after generating the question "Who are the presidents of France?" and retrieving the full answer subgraph from the KG (which would be now guaranteed to contain all presidents of France, even those that were not present in the original subgraph seen by the LLM in its prompt), this subgraph could be fed back to the LLM to generate questions like "How many presidents did France have?". We plan to look more into such promising additions to the SynthKGQA pipeline in future work.
> > >
> > > >2. Applicability of "High Recall / Low Precision" Strategy to Smaller Models
> > >
> > > We recognize the importance of considering different ranges of model sizes (and capabilities) in the analysis. We repeated the experiments with SubgraphRAG (varying the size of the retrieved subgraph) with LLama-3.1-8B and 70B, in addition to GPT-4o-mini and GPT-5-mini. We show the comparison in the new Figure C7 (where we aggregated all these results), and discuss it in the main text. Even for the 8B model there is no degradation in the final predictive accuracy when increasing the size of the retrieved subgraph up to 500 edges, showing that optimizing for recall is not detrimental, even though the performance eventually plateaus (so precision should not be entirely disregarded for more compact LLMs). The 70B model, on the other hand, already displays continued improvements up to 400+ edges. All this supports the conclusion that -- for more recent and capable LLMs, even open-sourced ones -- retrieval precision is likely becoming a less critical factor. Nevertheless, as stated before, we still hope to see the community develop new architectures for KG retrieval that exhibit good recall and good precision at the same time.
> > >
> > > As we show in the new Figure C7, LLMs of all sizes benefit from supervising training with the ground-truth triples in GTSQA, rather than using shortest paths; this improvement appears to become even larger for more capable LLMs.

---

> > > ### Author Response · Authors · 2025-11-26
> > > **Reply to reviewer vG9L (2/2)**
> > >
> > > >3. Quantification of Semantic Bias ("Jeopardy-style" Questions)
> > > 4. Lack of Human Verification for Quality Assurance
> > >
> > > To address the points raised by the reviewer, we now present 3 randomly sampled questions and their respective paraphrased version for every graph isomorphism type with at least 50 questions in the training dataset (Appendix A.4). We believe that, rather than trying to numerically quantify how natural/human-like the questions generated by our framework sound (which would inevitably result in a subjective evaluation), it is better to just show random examples to the reader.
> > > While we agree that there is no theoretical guarantee that the paraphrasing mitigates any linguistic bias originating from our data generation process, it can be seen in this sample that in practice it very often helps in mitigating structure biases and improving the naturalness of the questions (e.g. Question: "Which singer, born in Suphan Buri, holds citizenship of Thailand and plays the guitar?", Paraphrased Question: "Which Thai singer from Suphan Buri plays the guitar?").
> > >
> > > With respect to the reviewer's remarks on semantic focus of questions, we do not believe that the generation pipeline in our framework introduces more bias than previously-developed datasets, which often started from artificial templates for the logic queries. Since we are not generating the question directly from the answer subgraph, but we are providing a portion of the KG as context to the LLM and give it the freedom to build a question whose answer can be anywhere in it, we argue that we introduce even fewer constraints on the semantic focus of the resulting questions (the answer is equally likely to be either the head or the tail node of an edge -- i.e., either the subject or the attribute). As shown in the examples, we consistently see questions of the type auspicated by the reviewer, where the answer is the attribute of the relation encoded by an edge (e.g., "What was the military rank of Pierre Gaston-Mayer?"), and the paraphrasing also helps on this (e.g., Question: "Which resident of District 2 was killed by Thresh?", Paraphrased Question: "Who was the resident of District 2 that Thresh killed?"). It is true that especially complex questions which require multiple intersecting paths tend to follow the "Jeopardy-style" described by the reviewer, but we believe that this is inherent to this category of questions (as it is often observed --- to a much higher degree of artificiality -- for the more complex questions in datasets like CWQ and GrailQA), and it is does not constitute a limit to real-world utility, when trying to benchmark the abilities of KG retrievers.
> > >
> > > We hope that this manual exploration of questions of different complexity can also assure the reviewer and the general readership of the high overall quality of the questions generated by SynthKGQA.

---

### Author Response · Authors · 2025-12-03
**Rebuttal summary for the AC**

We again thank the reviewers for their valuable input that helped us to address gaps in our manuscript and improve the overall paper. We believe we have addressed all concerns and hope to convince the reviewers and the area chair of the value and the novelty of our contribution. In particular,
 - we resolved the concerns of reviewer vG9L regarding the significance of comparing the  retrievers trained on the ground-truth subgraphs provided by our SynthKGQA framework against the ones trained on the shortest path heuristic, and the statistical rigor of such comparison. We also extended our analysis to using GPT-4o-mini and GPT-5-mini as final LLM, with interesting new insights into the relevance of retrieval precision vs recall, convincing the reviewer to increase their score. As additionally requested by the reviewer, we furthermore included LLama-3.1-8B and LLama-3.1-70B as representatives of smaller open-source models in this analysis, to reinforce our findings on both the precision/recall tradeoff and the value of the ground-truth subgraphs provided by our framework.
- we added representative, randomly sampled questions and their paraphrased versions for every graph isomorphism type in Appendix A.4. This is to demonstrate that questions generated by SynthKGQA are of high-quality and also sound natural, to address concerns regarding any semantic bias in the generation framework, compared to previous approaches.
- we clearly laid out the shortcomings of prior and concurrently-developed frameworks such as ConvKGYarn, Dynamic-KGQA, KGQAGen and how SynthKGQA mitigates these. In a nutshell, SynthKGQA significantly improves on the factual correctness of generated datapoints and uniquely provides all necessary information (seed nodes, full set of answers and ground-truth answer subgraphs) to fully benchmark and better train KG retriever models. Furthermore, while all these other papers only focus on the data-generation framework (providing very limited experiments that use the generated data), we clarified to reviewers F3Vj and EYwG the many other novel contributions of our work, including:
    - a comprehensive investigation of the generalization abilities of trainable KG retrievers;
    - and in-depth discussion on the trade-off between retrieval recall and precision when KG retrievers are combined with SOTA LLMs (made possible by the complete ground-truth subgraphs provided by SynthKGQA), providing useful guidance to the community on how to develop better architectures for KG retrieval;
     - an extensive discussion on the shortcomings of many SOTA KG retrievers, that were not previously highlighted in the literature or detectable through previous benchmarks (as supported also by the leaderboard comparison provided in the reply to reviewer F3Vj);
    - the first quantitative assessment of the value of ground-truth subgraphs for training KG-retrievers, compared to using the commonly-adopted shortest path heuristic.

---

### Meta-Review · Area_Chair_Qh8b · 2026-01-07

**Summary:**

The reviewers agree that the paper addresses a relevant problem in KG-augmented LLMs and presents a technically sound framework for generating synthetic KBQA data with ground-truth subgraphs. However, multiple reviewers raise serious concerns about the paper’s novelty, external validity, and overall research contribution, which are not fully resolved by the rebuttal.

Several reviewers argue that the core ideas, including LLM-based question generation from subgraphs and SPARQL-based validation, substantially overlap with prior work, and that the paper does not clearly articulate what is fundamentally new beyond incremental refinements **[F3Vj, EYwG]**.

Other reviewers acknowledge the engineering effort but express reservations about the closed-loop evaluation setting, limited evidence of real-world relevance, and the narrow scope of the benchmark relative to practical KGQA needs **[vG9L, EYwG]**.

While the rebuttal adds clarifications and additional analysis, reviewers who were initially skeptical maintain that these changes do not materially strengthen the paper’s core claims **[F3Vj, EYwG, vG9L]**.

**Reviewer Concerns:**

**1. Novelty relative to prior and concurrent work**
Reviewers F3Vj and EYwG emphasize that key components of the framework have already been explored in earlier or concurrent work, and that the paper does not sufficiently distinguish its contributions from these efforts **[F3Vj, EYwG]**. These concerns remain after the rebuttal **[F3Vj]**

**2. Closed-loop evaluation and external validity**
Reviewer vG9L highlights that the experiments primarily demonstrate synthetic-to-synthetic gains, without evidence of improved performance on established, human-created KGQA benchmarks. The rebuttal explains why such comparisons were not performed, but the reviewer maintains that this limits practical relevance.

**3. Scope and task coverage limitations**
Reviewers note that the benchmark focuses mainly on path-based retrieval questions and excludes aggregation and other non-retrieval logic common in real-world KGQA, potentially narrowing its applicability **[vG9L, F3Vj]**.

**4. Strength of empirical evidence**
Concerns are raised that some empirical gains rely on comparisons against weak or well-known baselines, and that improvements may partly reflect dataset construction choices rather than fundamentally stronger training signals **[vG9L, F3Vj]**.

**Reviewer Scores:**

Reviewer F3Vj would likely keep their score unchanged, as they explicitly state that their concerns about novelty and contribution remain.

Reviewer EYwG would likely keep their score unchanged, as the rebuttal does not materially alter their assessment of overlap with prior work.

Reviewer vG9L increased their score after the rebuttal but continues to emphasize unresolved concerns about external validity and scope, suggesting no further upward change.

Overall, the discussion does not indicate a shift toward acceptance.

---

### Decision · Program_Chairs · 2026-01-26

Reject